

# Necklace Ansatz for strongly repulsive spin mixtures on a ring

Gianni Aupetit-Diallo[1⋆], Giovanni Pecci[1†], Artem Volosniev[2‡],
Mathias Albert[3,4∘], Anna Minguzzi[5§] and Patrizia Vignolo[3,4¶]

**1** SISSA, Via Bonomea 265, I-34136 Trieste, Italy
**2** Department of Physics and Astronomy, Aarhus University,
Ny Munkegade 120, DK-8000 Aarhus C, Denmark
**3** Université Côte d'Azur, CNRS, Institut de Physique de Nice, 06200 Nice, France
**4** Institut Universitaire de France
**5** Université Grenoble Alpes, CNRS, LPMMC, 38000 Grenoble, France

⋆ gaupetit@sissa.it , † gpecci@sissa.it , ‡ artem@phys.au.dk ,
∘ mathias.albert@univ-cotedazur.fr , § anna.minguzzi@lpmmc.cnrs.fr ,
¶ patrizia.vignolo@univ-cotedazur.fr

## Abstract

We propose an alternative to the Bethe Ansatz method for repulsive strongly-interacting fermionic (or bosonic) mixtures on a ring. Starting from the knowledge of the solution for single-component non-interacting fermions (or strongly-interacting bosons), we explicitly impose periodic condition on the amplitudes of the spin configurations. This reduces drastically the number of independent complex amplitudes that we determine by constrained diagonalization of an effective Hamiltonian. This procedure allows us to obtain a complete basis for the exact low-energy many-body solutions for mixtures with a large number of particles, both for $SU(\kappa)$ and symmetry-breaking systems.

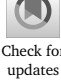

## 1  Introduction

Exactly solvable quantum many-body systems are rare in physics, and generally exist only in one-dimensional (1D) spatial geometries. For a long time these systems have been seen more as toy models rather than models that can describe real physical set-ups. However, in the last decades they gained also this status thanks to their implementation in cold-atom laboratories that enabled quantum simulations of many-body [1–3] as well as of few-body physics in 1D [4–6].

Ultracold gases are extremely rich and versatile. They can be realized with bosonic or fermionic atoms, which can be non-interacting or with tunable interactions up-to very strong repulsive or attractive interactions, and eventually with a spin or color degree of freedom that can be very large [7]. The stronger the interaction strength, the more correlated the atoms are, and the more difficult it is to get numerically an accurate description of the system, especially for long-time dynamics. For these reasons, exact solutions for quantum systems are gaining more and more interest and are becoming essential both for deep understanding of fundamental physics and for benchmarking classical and quantum simulators.

Exact solutions for 1D homogeneous systems are well-known in the literature. Celebrated examples are 1D bosons or fermions with contact interactions that are solvable by the Bethe Ansatz both in the case of repulsions [8–12] and attractions [3, 13–18], the latter giving rise to many-body bound states or pairing. In the presence of an inhomogeneous confinement there are generally no exact solutions for interacting systems, except for systems with infinite repulsive interactions such as impenetrable bosons, the Tonks-Girardeau (TG) gas [19], or impenetrable bosonic or fermionic mixtures [2, 4, 5, 20–23]. The key point for this category of exact solutions is that impenetrable particles behave like non-interacting fermions as long as the correct symmetry exchange is taken into account. The mixture many-body wavefunction is thus mapped on that for non-interacting fermions, and the exchange properties are determined by the diagonalization of an effective Hamiltonian related to the contact matrix [21, 23–26].

Until now, this method was essentially applied only to mixtures under external confinement, such as harmonic or box traps, and not to ring systems, with the sole exception of the ground-state of non-degenerate symmetric mixtures with vanishing momentum [27]. Ring trapping potentials have become available in the experiments with ultracold atoms, and are nowadays realized with unprecedented precision and smoothness (see e.g. Ref. [28] and references therein). The ring geometry, corresponding to imposing periodic boundary conditions, is the most suitable geometry to study the thermodynamic limit, due to absence of boundaries or inhomogeneities. Also, finite-size rings are important for studying mesosocopic effects, such as the response to applied gauge fields [29,30] as well as for applications to atomtronics [31].

The exact solutions for mixtures on a ring are generally provided by the Bethe Ansatz [10, 11, 13, 32], whose resolution becomes increasingly complex as the number of atoms and components increases [33–37]. Inspired by the contact matrix method introduced in [21, 23], we propose here an alternative to the Bethe Ansatz, the necklace Ansatz, for calculating the spectrum and the many-body eigenstates for fermionic or bosonic mixtures in the strongly repulsive limit.

The first building block of our procedure is the solution for a single-component quantum gas on a ring. It allows us to write the many-body wavefunction of a fermionic/bosonic mixture for a given sector, i.e., a given order of particles. The second step consists in regrouping the sectors that are equivalent up to a permutation of identical particles in snippets. The number of snippets fixes the number of independent solutions for the spin components. The third step, which is the crucial point of the procedure, is to regroup the snippets that are the same on the ring up to a rotation, i.e. that belong to the same *necklace*. The fact that the many-body wavefunction has to be the same on snippets belonging at the same necklace fixes a phase relation between the snippets' amplitudes. This reduces further the complexity of the problem: one needs to determine a number of complex coefficients that is equal to the number of different necklaces minus one, because of the normalization condition. The last step is to determine these complex amplitudes. This is done by solving a constrained diagonalization of an effective Hamiltonian for each value of the quantized total momentum.

The article is organized as following. In Sec. 2 we remind the solution for a spinless Fermi gas and for a single-component TG gas on a ring and we remind the main steps of the Bethe Ansatz in order to find the solution for a fermionic/bosonic mixture in the strongly repulsive limit. The necklace Ansatz is detailed in Sec. 3 and many examples are given in Sec. 4. Our concluding remarks are given in Sec. 5.

## 2 Strongly-interacting quantum gases on a ring

### 2.1 Single-component particles on a ring

Let us consider the ground state for $N$ spinless fermions or $N$ TG bosons on a ring of length $L$, with coordinates $X = \{x_1, \dots, x_N\}$ [38, 39]:

$$\Psi_{GS}(X) = \begin{cases} \Psi_{SD}(k_\ell^F x_j), \\ \mathcal{A}\Psi_{SD}(k_\ell^B x_j), \end{cases} \tag{1}$$

with $\Psi_{SD}(k_\ell x_j)$ being the Slater determinant built with the ring single-particle orbital solutions $\sim e^{ik_\ell x_j}$ and $\mathcal{A}$ is the symmetrization operator.

For the case of $N$ even fermions, $k_\ell^F = \{-\frac{N}{2}\frac{2\pi}{L}, (-\frac{N}{2}+1)\frac{2\pi}{L}, \dots, 0, \dots, (\frac{N}{2}-1)\frac{2\pi}{L}\}$; for the case of $N$ odd fermions, $k_\ell^F = \{-\frac{N-1}{2}\frac{2\pi}{L}, \dots, 0, \dots, \frac{N-1}{2}\frac{2\pi}{L}\}$; for the case of $N$ even TG bosons, $k_\ell^B = \{-\frac{N-1}{2}\frac{2\pi}{L}, \dots, -\frac{\pi}{L}, +\frac{\pi}{L}, \dots, \frac{N-1}{2}\frac{2\pi}{L}\}$; and for the case of $N$ odd TG bosons, $k_\ell^B = \{-\frac{N-1}{2}\frac{2\pi}{L}, \dots, 0, \dots, \frac{N-1}{2}\frac{2\pi}{L}\}$.

Remark that, for all $\mathcal{K} = \frac{2\pi n}{L}$,

$$\Psi_{\mathcal{K}}(X) = e^{i\mathcal{K}(\sum_j x_j)/N)}\Psi_{GS}(X) = \begin{cases} \Psi_{SD}((k_\ell^F + \mathcal{K}/N)x_j), \\ \mathcal{A}\Psi_{SD}((k_\ell^B + \mathcal{K}/N)x_j), \end{cases} \tag{2}$$

is a solution with total momentum

$$\mathcal{P}^{N,n} = \sum_{j=1}^{N} \hbar\left(k_j^{F,B} + \frac{\mathcal{K}}{N}\right) = \sum_{j=1}^{N} \hbar k_j^{F,B} + \frac{2n\pi\hbar}{L} = \mathcal{P}_{F,B}^{N,n} + \hbar\mathcal{K}, \tag{3}$$

and energy

$$E_\infty^{N,n} = \frac{\hbar^2}{2m}\sum_{j=1}^{N}\left(k_j^{F,B} + \frac{\mathcal{K}}{N}\right)^2 = \frac{\hbar^2}{2m}\sum_{j=1}^{N}\left(k_j^{F,B} + \frac{2n\pi}{LN}\right)^2 . \tag{4}$$

## 2.2 Quantum mixtures on a ring

Let us now consider a fermionic or a bosonic spin mixture with $\kappa$ components, obeying the Hamiltonian

$$\hat{H} = \sum_{\sigma=1}^{\kappa}\sum_{i=1}^{N_\sigma} -\frac{\hbar^2}{2m}\frac{\partial^2}{\partial x_{i,\sigma}^2} + \sum_{\sigma}^{\kappa} g_{\sigma\sigma}\sum_{i=1}^{N_\sigma}\sum_{j>i}^{N_\sigma}\delta(x_{i,\sigma}-x_{j,\sigma}) + \frac{1}{2}\sum_{\sigma\neq\sigma'=1}^{\kappa} g_{\sigma\sigma'}\sum_{i=1}^{N_\sigma}\sum_{j=1}^{N_{\sigma'}}\delta(x_{i,\sigma}-x_{j,\sigma'}), \tag{5}$$

where the $i,j$'s are the particle's indices that go from 1 to $N_\sigma$ ($N_{\sigma'}$), the $\sigma,\sigma'$'s are the spin indices that go from 1 to $\kappa$ and the $g_{\sigma\sigma'}$'s are the inter- and intra-species interaction strengths. The latter concerns only identical bosons since for identical fermions $s$-waves contact interactions are not allowed. Here and in the following, we are interested in the limit $g_{\sigma\sigma'} \to +\infty$, for any $\sigma,\sigma'$. In this strongly interacting regime, the many-body wavefunction vanishes whenever $x_i = x_j$.

## 2.3 The Bethe Ansatz solution at strong interaction

In this section, we briefly summarize the Bethe Ansatz solution for strongly-interacting quantum mixtures on a ring. In SU($\kappa$) mixtures, when $g_{\sigma\sigma'} = g$ for any $\sigma,\sigma'$, the systems described by Hamiltonian (5) can be solved exactly at any interaction strength and for generic $\kappa$ using the Bethe Ansatz [11,40–44].

In the following, we only treat the strongly interacting regime $g \to \infty$, as it constitutes the main focus of this work. In this regime, in each coordinate sector $\theta_Q(X) = \theta(x_{Q(1)} < x_{Q(2)}\cdots < x_{Q(N)})$, $Q$ being the permutation operator, the Bethe Ansatz wavefunction reads:

$$\Psi_{BA,Q}(X) = a_Q(\Lambda_1^{(2)},\ldots\Lambda_{N_2}^{(2)};\ldots;\Lambda_1^{(\kappa)},\ldots\Lambda_{N_\kappa}^{(\kappa)})\sum_P (-1)^{(1-\eta_B)|P|}\exp\left\{i\sum_j k_{P(j)}x_{Q(j)}\right\}, \tag{6}$$

where the wavevectors $k_j$, $j = 1\ldots N$ are the charge rapidities; $\Lambda_m^{(\sigma)}$ are the spin rapidities rescaled by the interaction constant [45,46] with $\sigma = 2\ldots\kappa$ being the label of the spin species and $m = 1\ldots N_\sigma$ being the label of the particle with the spin $\sigma$. The sum is performed over all the possible permutations $P$ in the symmetric group $S_N$; $\eta_B = 0,1$ for fermionic and bosonic mixtures, respectively. The notation $|P|$ indicates the number of transpositions linking the permutation $P$ with the identical sector defined by $k_1 \leq k_2 \leq \cdots \leq k_N$. We remark that, at intermediate or weak interactions, the amplitudes $a_Q$ depend also on the charge rapidities. As a consequence, in the general case, the amplitudes also depend on the permutation $P$. However, in the strongly interacting limit we consider, such dependence drops out due to the decoupling of spin and charge degrees of freedom, and the amplitudes of the Bethe wavefunction only depend on the permutation $Q$ indicating the coordinate sector.

The charge and spin rapidities are specified by a set of charge and a set of spin quantum numbers, respectively $\{\mathcal{I}_j\}$, $j = 1\ldots N$ and $\{\mathcal{J}_m^{(\sigma)}\}$, $m = 1\ldots N_\sigma$, and they are determined by imposing the periodic boundary conditions for the charge and for the spin part of the wavefunction [47,48].

In the strongly interacting regime, the charge rapidities can be calculated explicitly using:

$$Lk_j = 2\pi\left(\mathcal{I}_j \pm \frac{1}{N}\sum_{\sigma=2}^{\kappa}\sum_{n=1}^{N_\sigma}\mathcal{J}_m^{\sigma}\right), \tag{7}$$

where the $+$ sign is for fermions and the $-$ is for bosons. The possible values of the quantum numbers depend on the statistics of the particles. In the next paragraph, we outline the Bethe Ansatz solution for two-component mixtures at infinite interaction strength.

**SU(2) mixtures**  For the case $\kappa = 2$, the quantum numbers obey the following rules. For bosonic mixtures, $\{\mathcal{I}_j\}$ and $\{\mathcal{J}_m^{(2)}\}$ are integers if $N$ and $N_2$ have the same parity, and half-integers otherwise [44]. For Fermi gases, the nature of the quantum numbers is more complicated. For odd $N$, both $\{\mathcal{I}_j\}$ and $\{\mathcal{J}_m^{(2)}\}$ are integers or half-integers depending on $N_2$ being even or odd respectively. For even number of particles, $\{\mathcal{I}_j\}$ are integers and $\{\mathcal{J}_m^{(2)}\}$ are half-integers for even $N_2$, while for $N_2$ odd the quantum numbers are $\{\mathcal{I}_j\}$ and $\{\mathcal{J}_m^{(2)}\}$ are half-integers and integers respectively.

Bethe Ansatz provides an analytical expression for the amplitudes $a_Q(\Lambda_1^{(2)}, ... \Lambda_{N_2}^{(2)})$:

$$a_Q(\Lambda_1^{(2)}, ... \Lambda_{N_2}^{(2)}) \propto (-1)^{|Q|} \sum_R \prod_{1 \le m < n \le N_2} \frac{\Lambda_{R(m)}^{(2)} - \Lambda_{R(n)}^{(2)} - 2i}{\Lambda_{R(m)}^{(2)} - \Lambda_{R(n)}^{(2)}} \prod_{l=1}^{N_2} \left( \frac{\Lambda_{R(l)}^{(2)} - i}{\Lambda_{R(l)}^{(2)} + i} \right)^{y_{Q(l)}}, \qquad (8)$$

where the integer $y_{Q(l)}$ labels the position of the $l$-th spin down in the coordinate sector $\theta_Q(X)$ and $|Q|$ indicates the number of transpositions mapping $Q$ into the identical coordinate sector $x_1 \le x_2 \le \cdots \le x_N$. The summation runs over all the possible permutations $R$ of the $N_2$ spin rapidities.

For each set of $\{k_j\}$ and of spin quantum numbers $\{\mathcal{J}_m^{(2)}\}$, the spin rapidities can be obtained by imposing periodic boundary conditions on the amplitudes $a_Q$. This yields the $N_2$ *non-linear* coupled Bethe equations:

$$2N \arctan(\Lambda_m^{(2)}) = 2\pi \mathcal{J}_m^{(2)} + \sum_{n=1}^{N_2} 2 \arctan(\Lambda_m^{(2)} - \Lambda_n^{(2)}), \qquad m = 1 \dots N_2. \qquad (9)$$

In this limit, amplitudes (8) coincide, up to a normalization constant, with the ones of the Bethe wavefunction for the isotropic Heisenberg spin chain [49–53]. The Bethe equations (9) coincide with the ones of the Heisenberg model. This implies that the $1/g$-correction to the energy spectrum of the quantum mixture can be calculated from the Heisenberg spin chain with a suitable definition of an effective exchange coupling [46]. Remarkably, this equivalence holds for a generic value of $\kappa$ [54].

Finding all the solutions to the Bethe equations is a challenging task, already for moderate values of $N$ [33–35]. In order to obtain a complete set of complex roots [55] one has to include exceptional (or "singular") solutions and introduce regularizations of the equations [36, 56, 57]. Furthermore, the Bethe equations become significantly more intricate as $\kappa$ increases [37].

## 3   The necklace Ansatz

In this section we will outline an alternative procedure to the Bethe Ansatz at strong interaction for deriving the amplitudes $a_Q$ that we will call the necklace Ansatz. The word 'necklace' appears in combinatorics to describe a string of $N$ coloured beads, which can have up to $\kappa$ different colours, assuming that all rotations are equivalent [58].

Analogously to the trapped case [21, 23], in the fermionized regime, we can write the many-body wavefunction for a mixture starting from the single-component many-body wavefunction $\Psi_\mathcal{K}(X)$ (see Eq.(2)) on the basis of particle sectors $\theta_Q(X)$ yielding

$$\Psi(X) = \sum_{Q \in S_N} a_Q \theta_Q(X) \Psi_\mathcal{K}(X). \qquad (10)$$

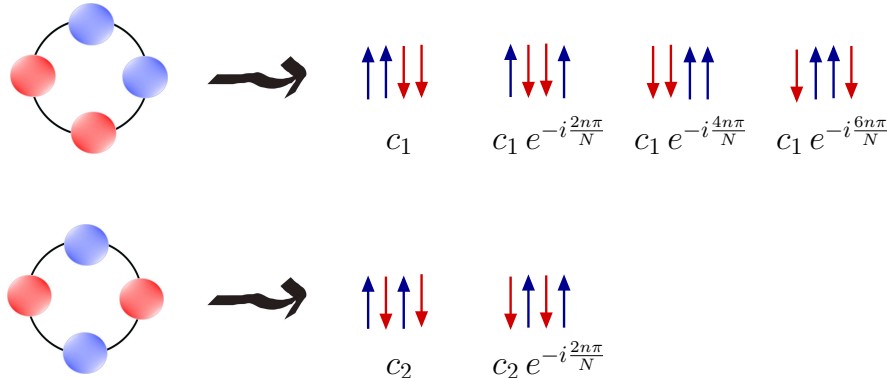

Figure 1: Schematic representation of the necklaces, their link to the snippets and the corresponding wavefunction amplitudes for the case of a 2+2 bosonic (fermionic) mixture.

The $N!$ sectors can be regrouped in $N_s = N!/(\prod_{\nu=1}^{\kappa} N_\nu!)$ snippets, with $N_s$ being the dimension of the Hilbert space, where each snippet is an ensemble of sectors that are equivalent under permutation of identical particles [5]. Notice that, for each snippet $s$, $a_Q = a_s$ for all $Q \in s$, using the symmetry under exchange of identical particles within the mixture. We can then rewrite Eq. (10) as

$$\Psi(X) = \sum_{s=1}^{N_s} a_s \Psi_{\mathcal{K},s}(X), \tag{11}$$

where $\Psi_{\mathcal{K},s}(X) = \sum_{Q \in s} \theta_Q(X) \Psi_{\mathcal{K}}(X)$ is the wavefunction that includes all coordinate sectors corresponding to the same snippet. We outline below the procedure for finding the $a_s$ coefficients for the $N_s$ independent spin-configurations corresponding to the ground- and excited states of the spin Hamiltonian.

In order to build the necklace Ansatz, the key observation is that, because of the periodic boundary conditions, there are families of equivalent snippets. This implies that the $a_s$ coefficients are not all independent. Let us clarify this point by making an example on two sectors for the sake of clarity. We consider the sector $x_1 < x_2 < x_3 < \cdots < x_N$ corresponding to the identity permutation $Q = \mathrm{Id}$, and the sector $x_2 < x_3 < \cdots < x_N < x_1$ obtained by the cyclic permutation $Q'(j) = j + 1$, where the cyclic condition implies $Q'(N) = 1$. On a ring these two sectors correspond – up to a rotation – to the same necklace, hence the many-body wavefunction has to be the same on these two sectors [49]. This means that, observing that the sector $\theta_{Q'}(x)$ can be obtained from $\theta_{\mathrm{Id}}(X)$ by applying the transformation $x_1 \to x_1 + L$, and taking into account that $\Psi_{\mathcal{K}}(x_1 + L, x_2, \ldots, x_N) = e^{i\mathcal{K}L/N}\Psi_{\mathcal{K}}(x_1, x_2, \ldots, x_N)$, the amplitudes in the sector $\theta_{Q'}(x)$ have to satisfy $a_{Q'} = e^{-i\mathcal{K}L/N} a_{\mathrm{Id}}$.

Using the fact that all the sectors that contribute to a given snippet have the same weight $a_Q$, we can extend the reasoning above to the snippets and regroup them in families when they correspond to the same necklace up-to a rotation (see Fig. 1). The $\mathcal{N}_\ell$ snippets, belonging to the same $\ell$-th necklace family, are connected by a $q$-cycle permutation $\tilde{Q}_q(i) = Q(i+q)$ with $q = 1, \ldots, \mathcal{N}_\ell - 1$. Setting $c_\ell$ the complex amplitude for a given snippet of the $\ell$-th necklace, the periodicity of the wavefunction along the ring imposes that in the many-body wavefunction the amplitudes of the snippets obtained by the $\tilde{Q}_q$ permutation coincide with $c_\ell$ times the phase factor $e^{-i\frac{q\mathcal{K}L}{N}}$, such that all the amplitudes of the snippets belonging to the $\ell$-th necklace read

$$c_\ell, \quad e^{-i\frac{\mathcal{K}L}{N}} c_\ell, \quad e^{-i\frac{2\mathcal{K}L}{N}} c_\ell, \quad \ldots, \quad e^{-i\frac{(\mathcal{N}_\ell - 1)\mathcal{K}L}{N}} c_\ell.$$

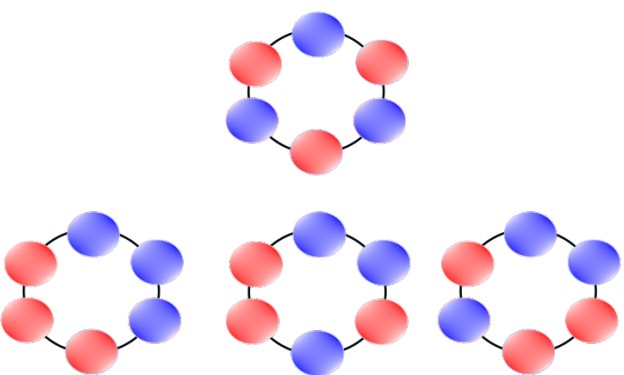

Figure 2: Necklaces for a 3+3 bosonic (fermionic) mixture. There is one high-symmetry necklace with $p_\ell = 2$ and three 6-period necklaces.

The number of snippets $\mathcal{N}_\ell$ belonging to the same necklace is generally $N$, except for high-symmetry necklaces that have a pattern that repeats itself with a period $p_\ell < N$, in such a case $\mathcal{N}_\ell = p_\ell$.

The condition for the $\ell$th rotated necklace to come to its initial position is that $\mathcal{N}_\ell \mathcal{K} L / N = 2\pi n$, with $n$ a relative integer. This recovers the condition $\mathcal{K} = 2\pi n / L$ for most families where $\mathcal{N}_\ell = N$, but gives a more restrictive condition for high-symmetry configurations. The allowed values of the quantum number $n$ depend on the type of necklace, see Sec. 4 for examples.

Notice that this Ansatz reduces drastically the number of coefficients required to define the wavefunction, from the number of snippets $N_s$ to the number of independent necklaces $N_{\text{neck}}$ (minus one, because of the normalization condition). The number of necklaces is given by

$$N_{\text{neck}} = (N_s - \sum_{j=1}^{M} p_j)/N + M \,, \tag{12}$$

$M$ being the number of high-symmetry configurations with period $p_\ell < N$. Notice that our Ansatz provides by construction a complete set of solutions at strong coupling. This can be a noticeably harder task if the Bethe Ansatz methods are used instead [34, 36, 55, 57, 59]. As an illustrative example, let us consider a two-component balanced mixture. For $N = 4$, $N_s = 6$ and $M = 1$ with period $p_1$ of length 2 ($\uparrow\downarrow\uparrow\downarrow$). This gives $N_{\text{neck}} = (6-2)/4+1 = 2$. Indeed we have two families of snippets that correspond to two necklaces (see Fig. 1): $\{\uparrow\uparrow\downarrow\downarrow, \uparrow\downarrow\downarrow\uparrow, \downarrow\downarrow\uparrow\uparrow, \downarrow\uparrow\uparrow\downarrow\}$ and $\{\uparrow\downarrow\uparrow\downarrow, \downarrow\uparrow\downarrow\uparrow\}$. For $N = 6$ we have $N_s = 20$ and again $M = 1$ with period of length $p_1 = 2$ ($\uparrow\downarrow\uparrow\downarrow\uparrow\downarrow$). This gives $N_{\text{neck}} = (6-2)/6+1 = 4$, see Fig. 2. For the case $N = 8$, $N_s = 70$. There are 2 possible periodic configurations with period smaller than $N$: one with period of length $p_1 = 2$, and the other with period of length $p_2 = 4$. This gives $N_{\text{neck}} = (70-2-4)/8+2 = 10$.

The coefficients in the wavefunction (10) are determined by solving the Schrödinger equation in the limit of large repulsive interactions. In this regime, spin and orbital part decouple and we are left to solve the eigenvalue problem for the contact matrix $V$ whose form depends on the type of mixture under consideration (see Appendices B and C for details and examples). In the fermionic $SU(2)$ case the contact matrix in the snippets' basis reads

$$[V^{SU^F}]_{i,j} = \frac{\hbar^4}{m^2} \begin{cases} \sum_{\ell_i} \delta_d \, \alpha_{\ell_i}, & j = i, \\ -v_{i,j}, & j \neq i, \end{cases} \tag{13}$$

while in the bosonic $SU(2)$ case we have

$$
[V^{SUB}]_{i,j} = \frac{\hbar^4}{m^2} \left\{ \begin{array}{ll} \sum_{\ell_i} (\delta_d + 2\delta_b) \alpha_{\ell_i}, & j = i, \\ v_{i,j}, & j \neq i, \end{array} \right. \tag{14}
$$

where $\delta_d$ is equal to one if $\ell_i$ and $\ell_i + 1$ correspond to distinguishable particles (two different spins) and zero otherwise, while $\delta_b$ is equal to one only for next-neighbor identical bosons. We indicate $\alpha_{\ell_i}$ as the nearest-neighbor exchange constant, given by

$$
\alpha_{\ell_i} = N! \int dx_1,\ldots dx_N \theta_{\mathrm{Id}}(x_1,\ldots,x_N) \delta(x_{\ell_i} - x_{\ell_i+1}) \left[ \frac{\partial \Psi_{\mathcal{K}}}{\partial x_{\ell_i}} \right]^2 . \tag{15}
$$

For the off-diagonal terms we have $v_{i,j} = \alpha_{\ell_i}$ if the snippets $i$ and $j$ differ from the exchange of two nearest-neighbor particles with different spins, set at positions $\ell_i$ ($\ell_j + 1$) and $\ell_i + 1$ ($\ell_j$), and zero otherwise.

In the homogeneous system, translation symmetry implies that all $\alpha_{\ell_i}$'s are equal and depend solely on the number of particles. From now on we will set $\alpha_{\ell_i} = \alpha^{(N)}$. We notice that the matrix $-V/g$ corresponds to the Hamiltonian of a Heisenberg spin chain with a hopping amplitude $\alpha^{(N)}/g$, that depends on the number of particles and interaction strength $g$ [21, 23–25, 27, 60].

## 4 Illustration of the method

In the next subsections, we will illustrate our method using a few representative cases. We show that the necklace Ansatz provides the same solutions as the Bethe Ansatz in the strongly interacting limit (see also results in Appendix D, whose range of validity is discussed in Appendix E). The illustrative examples below herald analysis of more complicated problems where the direct solution to the Bethe Ansatz equations is very difficult to access.

### 4.1 Spectrum and eigenstates of a 2+2 $SU(2)$ bosonic mixture

Let us consider the case of a bosonic SU(2) mixture with $N_\uparrow = 2$ $N_\downarrow = 2$ on a ring of length $L$. For such a system, the number of snippets is $N_s = 6$ and we have two necklaces, i.e. two families of snippets. To the snippets of the first necklace, given by $\{\uparrow\uparrow\downarrow\downarrow, \uparrow\downarrow\downarrow\uparrow, \downarrow\downarrow\uparrow\uparrow, \downarrow\uparrow\uparrow\downarrow\}$, we assign the coefficients $\{c_1, c_1 e^{-in\pi/2}, c_1 e^{-in\pi}, c_1 e^{-i3n\pi/2}\}$, while to the second necklace $\{\uparrow\downarrow\uparrow\downarrow, \downarrow\uparrow\downarrow\uparrow\}$, we assign the coefficients $\{c_2, c_2 e^{-in\pi/2}\}$, see Fig. 1. Remark that we have set $\mathcal{K}/N = 2n\pi/4$, $n$ being a relative integer, and that solutions with $c_2$ different from zero must have even values of $n$ ($c_2 e^{-in\pi} = c_2$). Thus, we expect two spin states if $n$ is even and only 1 when $n$ is odd and that Néel spin-configurations, namely configurations with alternating spin-up and spin-down, are forbidden in this case.

In order to find $c_1$ and $c_2$ as functions of $n$, we solve the conditioned eigenvalue problem

$$
\alpha^{(4)} \begin{pmatrix} 6 & 0 & 0 & 0 & 1 & 1 \\ 0 & 6 & 0 & 0 & 1 & 1 \\ 0 & 0 & 6 & 0 & 1 & 1 \\ 0 & 0 & 0 & 6 & 1 & 1 \\ 1 & 1 & 1 & 1 & 4 & 0 \\ 1 & 1 & 1 & 1 & 0 & 4 \end{pmatrix} \begin{pmatrix} c_1 \\ c_1 e^{-in\pi/2} \\ c_1 e^{-in\pi} \\ c_1 e^{-i3n\pi/2} \\ c_2 \\ c_2 e^{-in\pi/2} \end{pmatrix} = \xi_n \begin{pmatrix} c_1 \\ c_1 e^{-in\pi/2} \\ c_1 e^{-in\pi} \\ c_1 e^{-i3n\pi/2} \\ c_2 \\ c_2 e^{-in\pi/2} \end{pmatrix} . \tag{16}
$$

Setting $\tilde{\xi}_n = \xi_n/\alpha^{(4)}$, we obtain the following system of equations

$$
\begin{cases}
(6-\tilde{\xi}_n)c_1 & +(1+e^{-in\pi/2})c_2 & =0, \\
(6-\tilde{\xi}_n)c_1 e^{-in\pi/2} & +(1+e^{-in\pi/2})c_2 & =0, \\
(6-\tilde{\xi}_n)c_1 e^{-in\pi} & +(1+e^{-in\pi/2})c_2 & =0, \\
(6-\tilde{\xi}_n)c_1 e^{-i3n\pi/2} & +(1+e^{-in\pi/2})c_2 & =0, \\
(1+e^{-in\pi/2}+e^{-in\pi}+e^{-i3n\pi/2})c_1 & +(4-\tilde{\xi}_n)c_2 & =0, \\
(1+e^{-in\pi/2}+e^{-in\pi}+e^{-i3n\pi/2})c_1 & +(4-\tilde{\xi}_n)c_2 e^{-in\pi/2} & =0,
\end{cases}
\tag{17}
$$

that has six solutions summarized in Table 1. Note that there are two families of equations in Eq. (17): the first 4 differ only by the phase factor of the first member, and the second 2 by the phase factor of the second member. As Eq. (17) is overdetermined, we conclude that for $n \neq 0$, the solution has either $c_1 = 0$ or $c_2 = 0$. Only if $n = 0$ (modulo 4), we can have a situation when both coefficients are non-vanishing. We notice that the solution of Eq.(17) for $c_1 = 0$ shows that $c_2$ is different from zero only if $(1+e^{-in\pi/2}) = 0$, implying that $n$ is even, as was anticipated above. Similarly, the solution of Eq. (17) for $c_2 = 0$ yields as possible values $n = 1, 2, 3$ (modulo 4), showing that the allowed values of $\mathcal{K}$ are all the multiples of $2\pi\hbar/L$. This is at the origin of the fractionalization of the period of persistent currents, see Sec. 4.4. It follows from the above structure of the solution that the $c_j$'s solution with a given $n$ is a solution also for $n' = n + 4$.

One could think that there are too many equations for determining $c_1$ and $c_2$, but the imposition of this sort of gauge invariance condition for the two coefficients is necessary in order to find and select all (and only) the physical solutions. Indeed if one tries to get $c_1$ and $c_2$ directly from the minimization of the energy one obtains the equations

$$
\begin{cases}
(12-4\tilde{\xi}_n)c_1 & +(1+2\cos(n\pi/2)+\cos(n\pi))c_2 & =0, \\
(1+2\cos(n\pi/2)+\cos(n\pi))c_1 & +(4-2\tilde{\xi}_n)c_2 & =0,
\end{cases}
\tag{18}
$$

that allows solutions that are not physical, such as $c_2 \neq 0$ for $n = 1$.

To ascertain the validity of these solutions, one can verify that they have the correct symmetry. This could be done using the natural representations of $SU(\kappa)$ (or equivalently $S_N$), the Young diagrams. These diagrams are collections of boxes that schematically represent the particle-exchange symmetry of a physical state. Precisely, boxes in line (resp. column) refer to symmetric (resp. anti-symmetric) exchanges so that line diagram ⬚⬚⬚ corresponds to a three-particles fully symmetric state and a column ⬚ to a three-particles fully anti-symmetric state.

All other configurations represent more exotic states with mixed symmetries, each corresponding to a different representation of $S_N$. The usual procedure to connect our physical states to these diagrams is through the use of class sum operators, verifying that they are eigenstates of the 2-cycle class-sum operator $\Gamma^{(2)} = \frac{1}{2}\sum_{i<j} P_{i,j}$, [61,62] where $P_{i,j}$ is the operator which permutes the $i$-th and $j$-th elements. $\Gamma^{(2)}$'s eigenvalues are directly connected to the irreducible representations of $SU(\kappa)$, and thus to the Young diagrams. Indeed the relation between the eigenvalues $\gamma^{(2)}$'s and a Young diagrams with a number of boxes $\mu_i$ at row $i$ is [63]

$$
\gamma^{(2)} = \frac{1}{2}\sum_i [\mu_i(\mu_i - 2i + 1)].
\tag{19}
$$

For $n = 0$ the total momentum $\mathcal{P}$ is zero and we find (i) the fully symmetric state ⬚⬚⬚ with $c_1 = c_2 = 1/\sqrt{6}$ with eigenvalue $\xi_0/\alpha^{(4)} = 8$, and (ii) the solution $c_1 = 1/(2\sqrt{3})$, $c_2 = -1/\sqrt{3}$ with $\xi_0/\alpha^{(4)} = 2$ that corresponds to the symmetry represented by the Young diagram ⬚⬚. For $n = \pm1$ ($\mathcal{P} = \pm2\pi\hbar/L$), we find the solutions (iii) and (iv) with amplitudes $c_1 = 1/\sqrt{2}$, $c_2 = 0$ and $\xi_{\pm1}/\alpha^{(4)} = 6$. Both have symmetry ⬚⬚. For $n = 2$, we find two solutions: (v) one

Table 1: Solution for the 2+2 SU(2) bosonic mixture. $\tilde{\xi}_n$ are the rescaled eigenvalues $\xi_n/\alpha^{(4)}$. The last column indicates the symmetry of the solution with the associated Young diagram (YD).

| $n$ | $\tilde{\xi}_n$ | $c_1$ | $c_2$ | $\gamma^{(2)}$ | YD |
|-----|------|-------|-------|------|----|
| -1 | 6 | $1/2$ | $0$ | 2 | ⊞⊞ |
| 0 | 8 | $1/\sqrt{6}$ | $1/\sqrt{6}$ | 6 | ⊞⊞⊞ |
|    | 2 | $1/(2\sqrt{3})$ | $-1/\sqrt{3}$ | 0 | ⊞ |
| 1 | 6 | $1/2$ | $0$ | 2 | ⊞⊞ |
| 2 | 6 | $1/2$ | $0$ | 0 | ⊞ |
|    | 4 | $0$ | $1/\sqrt{2}$ | 2 | ⊞⊞ |

with $c_1 = 1/2$, $c_2 = 0$ and $\xi_2/\alpha^{(4)} = 6$ (⊞), and the last one (vi) with $c_1 = 0$, $c_2 = 1/\sqrt{2}$ and $\xi_2/\alpha^{(4)} = 4$ (⊞⊞). The values for the spin-states coefficients are summarized in Table 1. These states constitute a complete and orthogonal basis for the spin configurations. One can readily check that the obtained values for the $c_j$ yield the same wavefunctions as those obtained from the Bethe Ansatz solution in the strongly interacting limit (see e.g. [49, 64]).

## 4.2  Spectrum and eigenstates of a 4+2 SU(2) fermionic mixture

Let us now consider a 4+2 SU(2) fermionic mixture. For such a system $N_s = 6!/(4!2!) = 15$ and $N_{\text{neck}} = (15-3)/6 + 1 = 3$. To the snippets of the first necklace $\{\uparrow\uparrow\uparrow\uparrow\downarrow\downarrow, \uparrow\uparrow\uparrow\downarrow\downarrow\uparrow, \uparrow\uparrow\downarrow\downarrow\uparrow\uparrow,$ $\uparrow\downarrow\downarrow\uparrow\uparrow\uparrow, \downarrow\downarrow\uparrow\uparrow\uparrow\uparrow, \downarrow\uparrow\uparrow\uparrow\uparrow\downarrow\}$ we assign the coefficients $\{c_1, c_1 e^{-in\pi/3}, c_1 e^{-i2n\pi/3}, c_1 e^{-in\pi}, c_1 e^{-i4n\pi/3},$ $c_1 e^{-i5n\pi/3}\}$, to the second $\{\uparrow\uparrow\uparrow\downarrow\uparrow\downarrow, \uparrow\uparrow\downarrow\uparrow\downarrow\uparrow, \uparrow\downarrow\uparrow\downarrow\uparrow\uparrow, \downarrow\uparrow\downarrow\uparrow\uparrow\uparrow, \uparrow\downarrow\uparrow\uparrow\uparrow\downarrow, \downarrow\uparrow\uparrow\uparrow\downarrow\uparrow\}$, we assign the coefficients $\{c_2, c_2 e^{-in\pi/3}, c_2 e^{-i2n\pi/3}, c_2 e^{-in\pi}, c_2 e^{-i4n\pi/3}, c_2 e^{-i5n\pi/3}\}$, and to the third $\{\uparrow\uparrow\downarrow\uparrow\uparrow\downarrow,$ $\uparrow\downarrow\uparrow\uparrow\downarrow\uparrow, \downarrow\uparrow\uparrow\downarrow\uparrow\uparrow\}$ we assign the coefficients $\{c_3, c_3 e^{-in\pi/3}, c_3 e^{-i2n\pi/3}\}$. Remark that we have set $\mathcal{K}/N = 2n\pi/6$, $n$ being a relative integer, and that solutions with $c_3$ different from zero must have even values of $n$ ($c_3 e^{-in\pi} = c_3$). Thus, we expect 3 possible spin configurations if $n$ is even and only 2 if $n$ is odd. Indeed, by solving the conditioned eigenvalue system (C.1), we find that every even value of $n$ allows for 3 solutions and every odd $n$ allows for 2 solutions. Therefore, there are 15 independent solutions for $n$ from $n = -2$ to $n = 3$, that are given in Table 2.

## 4.3  3+3 mixtures: from SU(2) to symmetry breaking mixtures

The necklace Ansatz can also be used to analyze systems with $g_{\sigma\sigma'} \neq g_{\sigma\sigma}$ in Eq. (5) as long as the system is strongly interacting. In this subsection, we consider arguably the simplest scenario. The generalization to other cases is straightforward just as in the trapped cases [24, 60] or for the problem in a ring with vanishing total momentum [27].

Here we will consider a $3 + 3$ mixture and we will compare the cases of a SU(2) fermionic mixture and of a SU(2) bosonic mixture $g_{\uparrow\uparrow} = g_{\downarrow\downarrow} = g_{\uparrow\downarrow}$, with a symmetry breaking (SB) case of a bosonic mixture where the SU(2) symmetry is explicitly broken ($1/g_{\uparrow\uparrow} = 1/g_{\downarrow\downarrow} = 0$ and $g_{\uparrow\downarrow}$ is large but finite) [27]. For a $3 + 3$ mixture there are 20 snippets and $N_{\text{neck}} = 4$ independent necklaces, one of which has a period of length 2. We assign to the snippets of the first necklace $\{\uparrow\uparrow\uparrow\downarrow\downarrow\downarrow, \uparrow\uparrow\downarrow\downarrow\downarrow\uparrow, \uparrow\downarrow\downarrow\downarrow\uparrow\uparrow, \downarrow\downarrow\downarrow\uparrow\uparrow\uparrow, \downarrow\downarrow\uparrow\uparrow\uparrow\downarrow, \downarrow\uparrow\uparrow\uparrow\downarrow\downarrow\}$ the amplitudes $\{c_1, c_1 e^{-i\phi}, c_1 e^{-2i\phi}, c_1 e^{-3i\phi}, c_1 e^{-4i\phi}, c_1 e^{-5i\phi}\}$; to those of the second necklace $\{\uparrow\uparrow\downarrow\uparrow\downarrow\downarrow, \uparrow\downarrow\uparrow\downarrow\downarrow\uparrow,$ $\downarrow\uparrow\downarrow\downarrow\uparrow\uparrow, \uparrow\downarrow\downarrow\uparrow\uparrow\downarrow, \downarrow\downarrow\uparrow\uparrow\downarrow\uparrow, \downarrow\uparrow\uparrow\downarrow\uparrow\downarrow\}$ the amplitudes $\{c_2, c_2 e^{-i\phi}, c_2 e^{-2i\phi}, c_2 e^{-3i\phi}, c_2 e^{-4i\phi}, c_2 e^{-5i\phi}\}$; to those of the third necklace $\{\uparrow\downarrow\uparrow\uparrow\downarrow\downarrow, \downarrow\uparrow\uparrow\downarrow\downarrow\uparrow, \uparrow\uparrow\downarrow\downarrow\uparrow\downarrow, \uparrow\downarrow\downarrow\uparrow\downarrow\uparrow, \downarrow\downarrow\uparrow\downarrow\uparrow\uparrow, \downarrow\uparrow\downarrow\uparrow\uparrow\downarrow,\}$ the amplitudes $\{c_3, c_3 e^{-i\phi}, c_3 e^{-2i\phi}, c_3 e^{-3i\phi}, c_3 e^{-4i\phi}, c_3 e^{-5i\phi}\}$; and finally to the necklace of period 2 $\{\uparrow\downarrow\uparrow\downarrow\uparrow\downarrow, \downarrow\uparrow\downarrow\uparrow\downarrow\uparrow\}$, $\{c_4, c_4 e^{-i\phi}\}$ where $\phi = 2n\pi/6$ with $n$ being a relative integer. From the

Table 2: Solution for the 4+2 SU(2) fermionic mixture. $\tilde{\xi}_n$ are the rescaled eigenvalues $\xi_n/\alpha^{(6)}$. The coefficients $c_j$ are not normalized. The last column indicates the symmetry of the solution with the associated Young diagram (YD).

| $n$ | $\tilde{\xi}_n$ | $c_1$ | $c_2$ | $c_3$ | $\gamma^{(2)}$ | YD |
|---|---|---|---|---|---|---|
| -2 | 3 | $e^{2i\pi/3}$ | 1 | $2e^{i\pi/3}$ | -9 | |
| | $(7-\sqrt{17})/2$ | $-(3+\sqrt{17})/4\,e^{2i\pi/3}$ | 1 | $(\sqrt{17}-1)/4\,e^{i\pi/3}$ | -5 | |
| | $(7+\sqrt{17})/2$ | $(-3+\sqrt{17})/4\,e^{2i\pi/3}$ | 1 | $(\sqrt{17}+1)/4\,e^{-i2\pi/3}$ | -5 | |
| -1 | 1 | $\sqrt{3}e^{-i\pi/6}$ | 1 | 0 | -9 | |
| | 5 | $e^{i5\pi/6}/\sqrt{3}$ | 1 | 0 | -5 | |
| 0 | 0 | 1 | 1 | 1 | -15 | |
| | $5-\sqrt{5}$ | $-(1+\sqrt{5})/4$ | $(-1+\sqrt{5})/4$ | 1 | -5 | |
| | $5+\sqrt{5}$ | $(-1+\sqrt{5})/4$ | $-(1+\sqrt{5})/4$ | 1 | -5 | |
| 1 | 1 | $\sqrt{3}e^{i\pi/6}$ | 1 | 0 | -9 | |
| | 5 | $e^{-i5\pi/6}/\sqrt{3}$ | 1 | 0 | -5 | |
| 2 | 3 | $e^{-2i\pi/3}$ | 1 | $2e^{-i\pi/3}$ | -9 | |
| | $(7-\sqrt{17})/2$ | $-(3+\sqrt{17})/4\,e^{-2i\pi/3}$ | 1 | $(\sqrt{17}-1)/4\,e^{-i\pi/3}$ | -5 | |
| | $(7+\sqrt{17})/2$ | $(-3+\sqrt{17})/4\,e^{-2i\pi/3}$ | 1 | $(\sqrt{17}+1)/4\,e^{i2\pi/3}$ | -5 | |
| 3 | 2 | 1 | 0 | 0 | -5 | |
| | 4 | 0 | 1 | 0 | -9 | |

condition that $c_4 e^{-2i\phi} = c_4$, we expect that $c_4$ has to be zero if $n$ is not zero or a multiple of 3. So we expect 4 solutions if $n = 0$ and if $n = 3$, and 3 solutions for the cases $n = -2, -1, 1$ and 2, namely 20 independent solutions.

The contact $V$ matrices corresponding to the three different cases are given in Appendix C. The resulting spin states are presented in Table 3 for the SU(2) fermionic mixture, in Table 4 for the SU(2) bosonic mixture, and in Table 5 for the SB bosons. Remark that the SB states for a given $n$ are very similar to the SU(2) fermionic ones for $n \pm 3$. This is due to the fact that this SB bosonic mixture can be seen as a SU(2) fermionic mixture up-to a symmetrization operation of the particles in each component [27].

### 4.4 Persistent current in a SU(3) fermionic mixture

We will consider here the case of three SU(3) fermions rotating in a ring of length $L$ with rotation frequency $\Omega$ [54, 64]. The Hamiltonian of the system reads

$$\mathcal{H} = \sum_{j=1}^{3} \frac{1}{2m}\left(p_j - \frac{m\Omega L}{2\pi}\right)^2 + g\sum_{j<\ell}\delta(x_j - x_\ell). \tag{20}$$

The effect of the rotation is to produce an artificial gauge field yielding an effective flux $\Phi = \Omega L^2/(2\pi)$ [28–31], which in turn induces a persistent current of particles in the ring. Such currents can be used for characterizing the different phases of the system [65–69]. In analogy with superconducting rings, the ground state energy and the current are periodic functions of the gauge flux, with a period that is defined as the quantum of flux of the particles [70].

Table 3: Solution for the 3+3 SU(2) fermionic mixture. $\tilde{\xi}_n$ are the rescaled eigenvalues $\xi_n/\alpha^{(6)}$. The coefficients $c_j$ are not normalized. The last column indicates the symmetry of the solution with the associated Young diagram (YD).

| $n$ | $\tilde{\xi}_n$ | $c_1$ | $c_2$ | $c_3$ | $c_4$ | $\gamma^{(2)}$ | YD |
|---|---|---|---|---|---|---|---|
| -2 | $\frac{1}{2}(7-\sqrt{17})$ | $\frac{e^{2i\pi/3}}{2}(3+\sqrt{17})$ | $e^{2i\pi/3}$ | 1 | 0 | -5 | |
| | $\frac{1}{2}(7+\sqrt{17})$ | $\frac{e^{-i\pi/3}}{2}(-3+\sqrt{17})$ | $e^{2i\pi/3}$ | 1 | 0 | -5 | |
| | 3 | 0 | $e^{-i\pi/3}$ | 1 | 0 | -9 | |
| -1 | 1 | $2e^{i\pi/3}$ | $e^{i\pi/3}$ | 1 | 0 | -9 | |
| | 4 | $e^{i2\pi/3}$ | $e^{i\pi/3}$ | 1 | 0 | -3 | |
| | 5 | 0 | $e^{-i2\pi/3}$ | 1 | 0 | -5 | |
| 0 | 0 | 1 | 1 | 1 | 1 | -15 | |
| | 6 | 0 | 1 | -1 | 0 | -3 | |
| | $5-\sqrt{5}$ | $-\frac{1}{2}(3+\sqrt{5})$ | 1 | 1 | $\frac{3}{2}(-1+\sqrt{5})$ | -5 | |
| | $5+\sqrt{5}$ | $\frac{1}{2}(-3+\sqrt{5})$ | 1 | 1 | $-\frac{3}{2}(1+\sqrt{5})$ | -5 | |
| 1 | 1 | $2e^{-i\pi/3}$ | $e^{-i\pi/3}$ | 1 | 0 | -9 | |
| | 4 | $e^{-i2\pi/3}$ | $e^{-i\pi/3}$ | 1 | 0 | -3 | |
| | 5 | 0 | $e^{i2\pi/3}$ | 1 | 0 | -5 | |
| 2 | $\frac{1}{2}(7-\sqrt{17})$ | $\frac{e^{-2i\pi/3}}{2}(3+\sqrt{17})$ | $e^{-2i\pi/3}$ | 1 | 0 | -5 | |
| | $\frac{1}{2}(7+\sqrt{17})$ | $\frac{e^{i\pi/3}}{2}(-3+\sqrt{17})$ | $e^{-2i\pi/3}$ | 1 | 0 | -5 | |
| | 3 | 0 | $e^{i\pi/3}$ | 1 | 0 | -9 | |
| 3 | 2 | 0 | 1 | 1 | 0 | -5 | |
| | 4 | -1 | 1 | -1 | -3 | -9 | |
| | $5-\sqrt{13}$ | $\frac{1}{2}(3+\sqrt{13})$ | 1 | -1 | $\frac{1}{2}(1-\sqrt{13})$ | -3 | |
| | $5+\sqrt{13}$ | $\frac{1}{2}(3-\sqrt{13})$ | 1 | -1 | $\frac{1}{2}(1+\sqrt{13})$ | -3 | |

Strong repulsive [54, 64] and attractive [18, 71, 72] interactions induce a fractionalization of the period of the persistent current. In the attractive case, this phenomenon is related to the formation of two-body and many-body bound states, respectively for fermionic and bosonic mixtures. The period is reduced by a factor equal to the number of particles forming the bound state. In the repulsive case, the period of the persistent current is reduced, both for bosons and fermions, by a factor equal to the number of particles in the mixture. In this interaction regime, the fractionalization is due to the formation of spin excitations in the ground state of the gas, occurring as one applies the gauge flux [45, 46, 54].

Here, we use the necklace Ansatz to compute the energy levels of the SU(3) fermionic mixture, at infinite and at large but finite repulsive interactions, as a function of an effective gauge flux $\Phi$. In this case the snippets can be divided into two necklaces, one $\{\bullet \circ \triangleright, \circ \triangleright \bullet, \triangleright \bullet \circ\}$ with the amplitudes $\{c_1, c_1 e^{-i2\pi n/3}, c_1 e^{-i4\pi n/3}\}$, and another $\{\circ \bullet \triangleright, \bullet \triangleright \circ, \triangleright \circ \bullet\}$ with the amplitudes $\{c_2, c_2 e^{-i2\pi n/3}, c_2 e^{-i4\pi n/3}\}$. The computed spin states and the corresponding eigenvalues are given in Table 6.

Table 4: Solution for the 3+3 SU(2) bosonic mixture. $\tilde{\xi}_n$ are the rescaled eigenvalues $\xi_n/\alpha^{(6)}$. The coefficients $c_j$ are not normalized. The last column indicates the symmetry of the solution with the associated Young diagram (YD).

| $n$ | $\tilde{\xi}_n$ | $c_1$ | $c_2$ | $c_3$ | $c_4$ | $\gamma^{(2)}$ | YD |
|---|---|---|---|---|---|---|---|
| -2 | $\frac{1}{2}(17-\sqrt{17})$ | $\frac{e^{-i\pi/3}}{2}(-3+\sqrt{17})$ | $e^{2i\pi/3}$ | 1 | 0 | 5 | |
| | $\frac{1}{2}(17+\sqrt{17})$ | $\frac{e^{i\pi/3}}{2}(3+\sqrt{17})$ | $e^{2i\pi/3}$ | 1 | 0 | 5 | |
| | 9 | 0 | $e^{-i\pi/3}$ | 1 | 0 | 9 | |
| -1 | 11 | $2e^{i\pi/3}$ | $e^{i\pi/3}$ | 1 | 0 | 9 | |
| | 8 | $e^{-i2\pi/3}$ | $e^{i\pi/3}$ | 1 | 0 | 3 | |
| | 7 | 0 | $-e^{i\pi/3}$ | 1 | 0 | 5 | |
| 0 | 12 | 1 | 1 | 1 | 1 | 15 | |
| | 6 | 0 | 1 | -1 | 0 | 3 | |
| | $7-\sqrt{5}$ | $\frac{1}{2}(-3+\sqrt{5})$ | 1 | 1 | $-\frac{3}{2}(1+\sqrt{5})$ | 5 | |
| | $7+\sqrt{5}$ | $-\frac{1}{2}(3+\sqrt{5})$ | 1 | 1 | $\frac{3}{2}(1-\sqrt{5})$ | 5 | |
| 1 | 11 | $2e^{-i\pi/3}$ | $e^{-i\pi/3}$ | 1 | 0 | 9 | |
| | 8 | $e^{i2\pi/3}$ | $e^{-i\pi/3}$ | 1 | 0 | 3 | |
| | 7 | 0 | $-e^{-i\pi/3}$ | 1 | 0 | 5 | |
| 2 | $\frac{1}{2}(17-\sqrt{17})$ | $\frac{e^{i\pi/3}}{2}(-3+\sqrt{17})$ | $e^{-2i\pi/3}$ | 1 | 0 | 5 | |
| | $\frac{1}{2}(17+\sqrt{17})$ | $\frac{e^{-i\pi/3}}{2}(3+\sqrt{17})$ | $e^{-2i\pi/3}$ | 1 | 0 | 5 | |
| | 9 | 0 | $e^{i\pi/3}$ | 1 | 0 | 9 | |
| 3 | 10 | 0 | 1 | 1 | 0 | 5 | |
| | 8 | -1 | 1 | -1 | -3 | 9 | |
| | $7-\sqrt{13}$ | $\frac{1}{2}(3-\sqrt{13})$ | 1 | -1 | $\frac{1}{2}(1+\sqrt{13})$ | 3 | |
| | $7+\sqrt{13}$ | $\frac{1}{2}(3+\sqrt{13})$ | 1 | -1 | $\frac{1}{2}(1-\sqrt{13})$ | 3 | |

Table 5: Solution for the 3+3 symmetry breaking bosonic mixture. $\tilde{\xi}_n$ are the rescaled eigenvalues $\xi_n/\alpha^{(6)}$. The coefficients $c_j$ are not normalized. The last column indicates the expectation value of the 2-cycle class-sum operator $\Gamma^{(2)}$.

| $n$ | $\tilde{\xi}_n$ | $c_1$ | $c_2$ | $c_3$ | $c_4$ | $\langle\Gamma^{(2)}\rangle$ |
|---|---|---|---|---|---|---|
| -2 | 1 | $2e^{-i\pi/3}$ | $e^{i2\pi/3}$ | 1 | 0 | 5 |
| | 4 | $e^{i2\pi/3}$ | $e^{i2\pi/3}$ | 1 | 0 | 5 |
| | 5 | 0 | $e^{-i2\pi/3}$ | 1 | 0 | 8 |
| -1 | $\frac{1}{2}(7-\sqrt{17})$ | $\frac{e^{-2i\pi/3}}{2}(3+\sqrt{17})$ | $e^{i\pi/3}$ | 1 | 0 | 4.78 |
| | $\frac{1}{2}(7+\sqrt{17})$ | $\frac{e^{i\pi/3}}{2}(-3+\sqrt{17})$ | $e^{i\pi/3}$ | 1 | 0 | 7.21 |
| | 3 | 0 | $e^{-i2\pi/3}$ | 1 | 0 | 5 |
| 0 | 2 | 0 | 1 | -1 | 0 | 3 |
| | 4 | 1 | 1 | 1 | -3 | 7 |
| | $5-\sqrt{13}$ | $-\frac{1}{2}(3+\sqrt{13})$ | 1 | 1 | $\frac{1}{2}(1-\sqrt{13})$ | 5.67 |
| | $5+\sqrt{13}$ | $\frac{1}{2}(-3+\sqrt{13})$ | 1 | 1 | $\frac{1}{2}(1+\sqrt{13})$ | 9.72 |
| 1 | $\frac{1}{2}(7-\sqrt{17})$ | $\frac{e^{2i\pi/3}}{2}(3+\sqrt{17})$ | $e^{-i\pi/3}$ | 1 | 0 | 4.78 |
| | $\frac{1}{2}(7+\sqrt{17})$ | $\frac{e^{-i\pi/3}}{2}(-3+\sqrt{17})$ | $e^{-i\pi/3}$ | 1 | 0 | 7.21 |
| | 3 | 0 | $e^{i2\pi/3}$ | 1 | 0 | 5 |
| 2 | 1 | $2e^{i\pi/3}$ | $e^{-i2\pi/3}$ | 1 | 0 | 5 |
| | 4 | $e^{-i2\pi/3}$ | $e^{-i2\pi/3}$ | 1 | 0 | 5 |
| | 5 | 0 | $e^{i2\pi/3}$ | 1 | 0 | 8 |
| 3 | 0 | -1 | 1 | - 1 | 1 | 4.2 |
| | 6 | 0 | 1 | 1 | 0 | 5 |
| | $5-\sqrt{5}$ | $\frac{1}{2}(3+\sqrt{5})$ | 1 | -1 | $\frac{3}{2}(-1+\sqrt{5})$ | 3.611 |
| | $5+\sqrt{5}$ | $-\frac{1}{2}(-3+\sqrt{5})$ | 1 | -1 | $-\frac{3}{2}(1+\sqrt{5})$ | 7.18 |

Table 6: Solution for 3 SU(3) particles. $\tilde{\xi}_n$ are the rescaled eigenvalues $\xi_n/\tilde{\alpha}^{(3)}$. The last column indicates the symmetry of the solution with the associated Young diagram (YD).

| $n$ | $\tilde{\xi}_n$ | $c_1$ | $c_2$ | $\gamma^{(2)}$ | YD |
|---|---|---|---|---|---|
| -1 | 3 | $1/\sqrt{2}$ | 0 | 0 | ⊞ |
|  | 3 | 0 | $1/\sqrt{2}$ | 0 | ⊞ |
| 0 | 6 | 1/2 | -1/2 | 3 | ⊞⊞⊞ |
|  | 0 | 1/2 | 1/2 | -3 | ⊟ |
| 1 | 3 | $1/\sqrt{2}$ | 0 | 0 | ⊞ |
|  | 3 | 0 | $1/\sqrt{2}$ | 0 | ⊞ |

The effect of the rotation enters in the energy landscape. Indeed the energy of the spin state $\eta$ with the eigenvalue $\tilde{\xi}_{n_\eta}$ and the quantum number $n_\eta$

$$E_\eta = \varepsilon^{N,n_\eta} - \frac{\tilde{\xi}_{n_\eta}}{g}\alpha^{(N)}, \tag{21}$$

with

$$\varepsilon^{N,n_\eta} = \frac{\hbar^2}{2m}\sum_{j=1}^{N}\left(k_j^{F,B} + \frac{2n_\eta\pi}{LN} - \frac{2\pi}{L}\tilde{\Phi}\right)^2, \tag{22}$$

where $\tilde{\Phi} = \Phi/\Phi_0$, $\Phi_0 = h/m$.

We plot in Fig. 3 the energy landscape for the cases $g \to \infty$ (thin lines) and $g = 100mL/\hbar^2$. At infinite interactions, we observe the expected $1/N$ fractionalization of the periodicity of the energy landscape [45, 46, 54, 64]. At finite interactions, the parabola branches that display the most symmetric spin configurations and are centered in $\tilde{\Phi} = 0, 1, \ldots$ have the largest energy corrections and thus correspond to the lowest energy (see Fig. 3), while the intermediate parabolas have a higher energy, as expected when fractionalization is

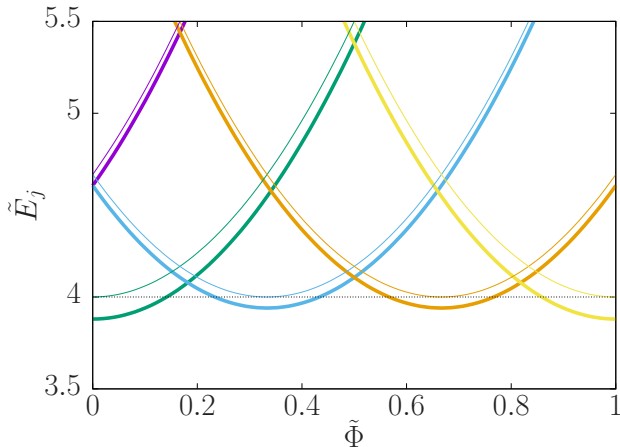

Figure 3: Energy landscape $\tilde{E}_j = E_j mL^2/(\hbar^2\pi^2)$ for a SU(3) three-fermions system, as a function of $\tilde{\Phi}$ at infinite interactions (thin lines) and for $g = 100mL/\hbar^2$ (thick lines). From left to right the different parabolas correspond to $n_j = -1$ (violet curves), 0 (green curves), 1 (blue curves), 2 (orange curves) and 3 (yellow curves). The black dotted horizontal line corresponds to the ground state energy of the three particles at $\tilde{\Phi} = 0$.

not yet achieved. This result displays the same features as those predicted in [46, 54] for the case of mixtures on a ring lattice: at increasing interactions, the parabolas corresponding to fractional values of the flux quantum decrease more and more in energy with respect to the ones centered at integers values of the reduced flux until they become all degenerate. To simplify the discussion, we have considered odd number of particles for each spin component. Conversely, we would have to include the parity effect, occurring in fermionic systems with even number of particles per spin species [70]. However, this does not change qualitatively the outcomes as it only results in a shift of the whole energy landscape by $\Phi_0/2$. We finally point out that, in the presence of a lattice, the energy correction depends also on the center-of-mass motion. In this case, not only the offset of the parabola branches is modified, but also their curvature [46, 54].

## 5  Concluding remarks

In this article, we have presented an alternative to the Bethe Ansatz, called the necklace Ansatz, which allows one to access the many-body wavefunction of a quantum mixture on a ring in the limit of strong repulsive contact interactions. Our method applies to both SU($\kappa$) mixtures and mixtures where the exchange symmetry is broken by taking different values for intra- and inter-component interaction strengths.

The necklace Ansatz allows one to obtain all the possible spin configurations very directly and brings a deep insight on their connection with the total momentum. The simplicity of this approach with respect to solving the Bethe's equations relies on the following facts: (i) being in the strongly interacting limit allows us to build the solution for an $N$-fermion (boson) mixture using as a building block the solution for $N$ free fermions ($N$ TG bosons); (ii) we reduce the dimension of the problem by organizing the sectors in snippets – groups of sectors that are equivalent under the permutation of identical particles, and the snippets in necklaces – groups of snippets that are equivalent up-to a rotation and thus can be represented by the same necklace. This drastically reduces the number of independent amplitudes to be determined in the many-body wavefunction. The remaining amplitudes, which are as many as the number of necklaces, are finally obtained by the constrained diagonalization of an effective Hamiltonian represented on the snippet basis. This approach yields by construction the complete basis of solutions in the given energy subspace. It also complements the method previously developed for mixtures under external confinement [21], solving the open issue on how to impose periodic boundary conditions within this formalism.

The necklace Ansatz tames the factorial increase of the Hilbert space with increasing the particle number. In addition, the connection between our Ansatz and combinatorial necklaces can mitigate the problem of generating and enumerating [58, 73, 74] possible orderings of strongly interacting atoms on a ring as well as spin configurations of the dual Heisenberg chain.[1] This will further simplify numerical calculations and allow one to obtain the many-body wavefunction of strongly correlated mixtures of relatively large systems. Such a wavefunction will be used as a starting point in the future for accurate calculations of both equilibrium and dynamical properties of 1D mixtures in ring geometries.

---

[1]Furthermore, this connection might guide the studies of mathematical symmetries of a few-body system (see, e.g., Refs. [75, 76] for corresponding studies in a trap) using the existing literature, see, e.g., Ref. [77].

## Acknowledgments

We thank Luigi Amico, Wayne Chetcuti, Nathan Harshman and Véronique Terras for enlightening discussions.

**Funding information** We acknowledge funding from the ANR-21-CE47-0009 Quantum-SOPHA project, and the support of the Institut Henri Poincaré (UAR 839 CNRS-Sorbonne Université), and LabEx CARMIN (ANR-10-LABX-59-01).

**Author contribution** GP and GAD have contributed equally to this work.

## A  Alternative derivation of the necklace Ansatz

For completeness, we present an alternative derivation of the Ansatz introduced in the main text. To this end, we study the auxiliary Hamiltonian

$$
\hat{h} = -\frac{\hbar^2}{2m}\frac{\partial^2}{\partial y} + \sum_{\sigma=1}^{\kappa}\sum_{i=1}^{N_\sigma} g_{I\sigma}\delta(y - x_{i,\sigma}) - \sum_{\sigma=1}^{\kappa}\sum_{i=1}^{N_\sigma}\frac{\hbar^2}{2m}\frac{\partial^2}{\partial x_{i,\sigma}^2}
$$
$$
+ \sum_{\sigma=1}^{\kappa} g_{\sigma\sigma}\sum_{i=1}^{N_\sigma}\sum_{j>i}^{N_\sigma}\delta(x_{i,\sigma} - x_{j,\sigma}) + \frac{1}{2}\sum_{\sigma\neq\sigma'=1}^{\kappa} g_{\sigma\sigma'}\sum_{i=1}^{N_\sigma}\sum_{j=1}^{N_{\sigma'}}\delta(x_{i,\sigma} - x_{j,\sigma'}). \tag{A.1}
$$

Here, we consider $N = \sum_{\sigma=1}^{\kappa} N_\sigma + 1$ particles, introducing explicitly a single 'impurity' atom ($y$ coordinate) in the Hamiltonian $\hat{h}$ in comparison to Eq. (5). Note that with a proper redefinition of the coordinates and the numbers of particles, the Hamiltonian $\hat{H}$ can always be written in the form of Eq. (A.1). By explicitly introducing the 'impurity' in this way, we set a reference frame, which allows us to order the snippets in a natural way.

Any solution to Eq. (A.1) has the form (see, e.g., Ref. [78])

$$
\Psi_h(y,X) = e^{i\mathcal{P}y/\hbar}\phi_{\mathcal{P}}(\tilde{X}), \tag{A.2}
$$

where $\tilde{X}$ is a set of coordinates of the majority particles measured with respect to the position of the impurity $y$, i.e., $\tilde{x}_{i,\sigma} = L\theta(y - x_{i,\sigma}) + x_{i,\sigma} - y$; $\mathcal{P}$ is the total momentum of the system. In the impenetrable limit, $1/g = 0$ and $1/g_{\sigma\sigma'} = 0$, the function $\phi_{\mathcal{P}}$ is an eigenstate of the Hamiltonian

$$
\hat{h}_{\mathcal{P}} = -\frac{\hbar^2}{2m}\sum_{\sigma=1}^{\kappa}\sum_{i=1}^{N_\sigma}\frac{\partial^2}{\partial\tilde{x}_{i,\sigma}^2} - \frac{\hbar^2}{2m}\left(\sum_{\sigma=1}^{\kappa}\sum_{i=1}^{N_\sigma}\frac{\partial}{\partial\tilde{x}_{i,\sigma}}\right)^2 + i\frac{\hbar\mathcal{P}}{m}\sum_{\sigma=1}^{\kappa}\sum_{i=1}^{N_\sigma}\frac{\partial}{\partial\tilde{x}_{i,\sigma}}. \tag{A.3}
$$

The last term in $\hat{h}_{\mathcal{P}}$ can be eliminated by the gauge transformation:

$$
\phi_{\mathcal{P}}(\tilde{X}) = \exp\left[i\frac{\mathcal{P}}{\hbar}\frac{\sum_{i,\sigma}\tilde{x}_{i,\sigma}}{1 + \sum_\sigma N_\sigma}\right]f_{\mathcal{P}}(\tilde{X}), \tag{A.4}
$$

where $f_{\mathcal{P}}$ solves the following Schrödinger equation

$$
\left[-\frac{\hbar^2}{2m}\sum_{\sigma=1}^{\kappa}\sum_{i=1}^{N_\sigma}\frac{\partial^2}{\partial\tilde{x}_{i,\sigma}^2} - \frac{\hbar^2}{2m}\left(\sum_{\sigma=1}^{\kappa}\sum_{i=1}^{N_\sigma}\frac{\partial}{\partial\tilde{x}_{i,\sigma}}\right)^2\right]f_{\mathcal{P}} = \epsilon f_{\mathcal{P}}. \tag{A.5}
$$

First, we note that for every ordering $Q(\tilde{X})$, $f_{\mathcal{P}}$ can be written as

$$f_{\mathcal{P}}(Q(\tilde{X})) = \tilde{c}_Q \exp\left[-i\frac{\mathcal{P}_F y}{\hbar} - i\frac{\mathcal{P}_F}{\hbar}\frac{\sum_{i,\sigma}\tilde{x}_{i,\sigma}}{1 + \sum_\sigma N_\sigma}\right]\Psi_F, \tag{A.6}$$

where $\Psi_F$ is a state that describes a system of $1 + \sum_\sigma N_\sigma$ spin-polarized fermions, and $\mathcal{P}_F$ is the total momentum of that state; $\tilde{c}_Q$ is an arbitrary coefficient. Indeed, the solution in Eq. (A.6) satisfies the boundary conditions, i.e., it vanishes whenever any two particles meet. To argue that any solution of Eq. (A.5) has the form of Eq. (A.6), one can design a proof by contradiction, i.e., show that one can construct a fermionic state $\Psi_F$ from a solution Eq. (A.5). To this end, it is useful to consider the known solution to the problem of one impurity in a Fermi gas in the frame co-moving with the impurity [79–81].

We see that any eigenstate of $\hat{h}$ has the form

$$\Psi_h(y,X) = \sum_Q \tilde{c}_Q \Theta_Q(\tilde{X}) \exp\left[i\frac{(\mathcal{P}-\mathcal{P}_F)y}{\hbar} + i\frac{\mathcal{P}-\mathcal{P}_F}{\hbar}\frac{\sum_{i,\sigma}\tilde{x}_{i,\sigma}}{1 + \sum_\sigma N_\sigma}\right]\Psi_F(y,X), \tag{A.7}$$

where the sum runs over all orderings of $\tilde{X}$ coordinates. The connection of this expression to the Ansatz in Eq. (10) becomes clear when working with the sector basis and remarking that $\mathcal{K} = (\mathcal{P}-\mathcal{P}_F)/\hbar$. Indeed, let us consider an example: three SU(3) fermions, i.e., the system in Sec. 4.4 with $\Phi = 0$. Without loss of generality, we assign the coordinates $y, x_1, x_2$ to the particles $\bullet \circ \triangleright$. Let us now write the function $\Psi_h(y,X)$ on the six possible sectors

$$(\bullet \circ \triangleright) = \{y < x_1 < x_2; \quad \tilde{x}_1 < \tilde{x}_2\}, \qquad \Psi_h(y,X) = \tilde{c}_1 \Psi_{\mathcal{K}}(y,X), \tag{A.8}$$

$$(\bullet \triangleright \circ) = \{y < x_2 < x_1; \quad \tilde{x}_2 < \tilde{x}_1\}, \qquad \Psi_h(y,X) = \tilde{c}_2 \Psi_{\mathcal{K}}(y,X), \tag{A.9}$$

$$(\triangleright \bullet \circ) = \{x_2 < y < x_1; \quad \tilde{x}_1 < \tilde{x}_2\}, \qquad \Psi_h(y,X) = \tilde{c}_1 e^{\frac{i\mathcal{K}L}{3}}\Psi_{\mathcal{K}}(y,X), \tag{A.10}$$

$$(\circ \bullet \triangleright) = \{x_1 < y < x_2; \quad \tilde{x}_2 < \tilde{x}_1\}, \qquad \Psi_h(y,X) = \tilde{c}_2 e^{\frac{i\mathcal{K}L}{3}}\Psi_{\mathcal{K}}(y,X), \tag{A.11}$$

$$(\circ \triangleright \bullet) = \{x_1 < x_2 < y; \quad \tilde{x}_1 < \tilde{x}_2\}, \qquad \Psi_h(y,X) = \tilde{c}_1 e^{\frac{2i\mathcal{K}L}{3}}\Psi_{\mathcal{K}}(y,X), \tag{A.12}$$

$$(\triangleright \circ \bullet) = \{x_2 < x_1 < y; \quad \tilde{x}_2 < \tilde{x}_1\}, \qquad \Psi_h(y,X) = \tilde{c}_2 e^{\frac{2i\mathcal{K}L}{3}}\Psi_{\mathcal{K}}(y,X), \tag{A.13}$$

where $\Psi_{\mathcal{K}}(y,X) = e^{i\mathcal{K}(y+x_1+x_2)/3}\Psi_F(y,X)$. To write down the wavefunction on sectors in this way, we used the definition of $\tilde{x}_i = L\theta(y-x_i) + x_i - y$ to connect the two orderings of $\{\tilde{x}_1, \tilde{x}_2\}$ to the six orderings of $\{y, x_1, x_2\}$. We see that $\Psi_h$ indeed reproduces the necklace Ansatz reported in Sec. 4.4 when choosing $\mathcal{K} = 2\pi n/L$, $c_1 = \tilde{c}_1$ and $c_2 = \tilde{c}_2 e^{2\pi n i/3}$.

Note that if the impurity belongs to the component $\sigma'$, by defining $Q' = Q(y \to x_{1,\sigma'})$, the values of $\tilde{c}_Q$ must satisfy the property: $\tilde{c}_Q = \tilde{c}_{Q'}\exp\left[i\frac{\mathcal{P}-\mathcal{P}_F}{\hbar(1+\sum_\sigma N_\sigma)}Lj_{y,x_{1,\sigma'}}\right]$, where an integer $j_{y,x_{1,\sigma'}}$ is the distance between $y$ and $x_{1,\sigma'}$ on a lattice for the ordering $Q$.

Finally, let us consider the system in Eq. (A.1) coupled to the Aharonov-Bohm flux as in Sec. 4.4. To this end, we substitute: $-i\hbar\partial/(\partial x_{i,\sigma}) \to -i\hbar\partial/(\partial x_{i,\sigma}) - m\Omega L/(2\pi)$ and for the impurity $-i\hbar\partial/(\partial y) \to -i\hbar\partial/(\partial y) - m\Omega L/(2\pi)$ in Eq. (A.1). The transformation to the co-moving with the 'impurity' frame of reference according to Eq. (A.2) separates the flux from the interaction potential explicitly, see, e.g., Supplementary of Ref. [82]. This means that the relative dynamics of the system is independent of the value of $\Phi$. In particular, the interaction between particles is disconnected from $\Phi$ in agreement with Sec. 4.4.

## B  Derivation of the effective Hamiltonian

In order to obtain the effective low-energy Hamiltonian, we have followed the procedure outlined in [23]. We expand in order of $1/g_{\sigma\sigma'}$ the Hamiltonian (5) on the snippet basis $\Psi_{\mathcal{K},s}$:

$$[\hat{H}]_{s,s'} \simeq [\hat{H}_{g_{\sigma\sigma'}\to\infty}]_{s,s'} + [H_{\text{eff}}]_{s,s'}, \tag{B.1}$$

where $[\hat{H}_{g_{\sigma\sigma'}\to\infty}]_{s,s'} = E_{\infty}^{N,n_s}\delta_{s,s'}$,

$$[H_{\text{eff}}]_{s,s'} = -\sum_{\sigma}^{\kappa}\frac{1}{g_{\sigma\sigma}}\left[\int dX \Psi_{\mathcal{K},s}^{*} g_{\sigma\sigma}^2 \sum_{i=1}^{N_\sigma}\sum_{j>i}^{N_\sigma}\delta(x_{i,\sigma}-x_{j,\sigma})\Psi_{\mathcal{K},s'}\right]_{g_{\sigma\sigma}\to\infty}$$
$$-\sum_{\sigma\neq\sigma'=1}^{\kappa}\frac{1}{g_{\sigma\sigma'}}\left[\frac{1}{2}\int dX \Psi_{\mathcal{K},s}^{*} g_{\sigma\sigma'}^2 \sum_{i=1}^{N_\sigma}\sum_{j=1}^{N_{\sigma'}}\delta(x_{i,\sigma}-x_{j,\sigma'})\Psi_{\mathcal{K},s'}\right]_{g_{\sigma\sigma'}\to\infty}. \tag{B.2}$$

By setting $g_{\sigma\sigma'} = \beta_{\sigma\sigma'}g$, we can write

$$[H_{\text{eff}}]_{s,s'} = -\frac{1}{g}[V]_{s,s'}, \tag{B.3}$$

where $V$ is the contact matrix, whose elements

$$[V]_{s,s'} = \sum_{\sigma}^{\kappa}\frac{1}{\beta_{\sigma\sigma}}\left[\int dX \Psi_{\mathcal{K},s}^{*} g_{\sigma\sigma}^2 \sum_{i=1}^{N_\sigma}\sum_{j>i}^{N_\sigma}\delta(x_{i,\sigma}-x_{j,\sigma})\Psi_{\mathcal{K},s'}\right]_{g_{\sigma\sigma}\to\infty}$$
$$+\sum_{\sigma\neq\sigma'=1}^{\kappa}\frac{1}{\beta_{\sigma\sigma'}}\left[\frac{1}{2}\int dX \Psi_{\mathcal{K},s}^{*} g_{\sigma\sigma'}^2 \sum_{i=1}^{N_\sigma}\sum_{j=1}^{N_{\sigma'}}\delta(x_{i,\sigma}-x_{j,\sigma'})\Psi_{\mathcal{K},s'}\right]_{g_{\sigma\sigma'}\to\infty}, \tag{B.4}$$

can be evaluated using the cusp condition [83].

For SU(2) mixtures $\beta_{\sigma\sigma'}$ is the same for any $\sigma,\sigma'$, so that we can set $\beta_{\sigma\sigma'} = 1$. We thus obtain Eqs. (13) and (14 respectively for fermions and bosons.

Remark that the contact matrix elements (15) do not depend on the momentum $\mathcal{K}$ $(n)$ [83]. Indeed $\alpha^{(N)}$ is equal, up to the dimensional constant $\hbar^2/(mL)$, to the difference between the total kinetic energy and the center-of-mass kinetic energy [27,84], thus it does not depend on the ring current [83]. This result is consistent with the Bethe formalism, by doing the expansion of the Bethe equations with respect to the inverse of the interaction strength (see Appendix D).

## C  Contact matrices

In this Appendix, we provide the contact matrices $V$ that have been used in order to obtain the results outlined in the main text.

The conditioned eigenvalues problem for the 4+2 SU(2) fermionic mixture takes the form

$$
\begin{pmatrix}
2 & 0 & 0 & 0 & 0 & 0 & -1 & 0 & 0 & 0 & 0 & -1 & 0 & 0 & 0 \\
0 & 2 & 0 & 0 & 0 & 0 & -1 & -1 & 0 & 0 & 0 & 0 & 0 & 0 & 0 \\
0 & 0 & 2 & 0 & 0 & 0 & 0 & -1 & -1 & 0 & 0 & 0 & 0 & 0 & 0 \\
0 & 0 & 0 & 2 & 0 & 0 & 0 & 0 & -1 & -1 & 0 & 0 & 0 & 0 & 0 \\
0 & 0 & 0 & 0 & 2 & 0 & 0 & 0 & 0 & -1 & -1 & 0 & 0 & 0 & 0 \\
0 & 0 & 0 & 0 & 0 & 2 & 0 & 0 & 0 & 0 & -1 & -1 & 0 & 0 & 0 \\
-1 & -1 & 0 & 0 & 0 & 0 & 4 & 0 & 0 & 0 & 0 & 0 & -1 & 0 & -1 \\
0 & -1 & -1 & 0 & 0 & 0 & 0 & 4 & 0 & 0 & 0 & 0 & -1 & -1 & 0 \\
0 & 0 & -1 & -1 & 0 & 0 & 0 & 0 & 4 & 0 & 0 & 0 & 0 & -1 & -1 \\
0 & 0 & 0 & -1 & -1 & 0 & 0 & 0 & 0 & 4 & 0 & 0 & -1 & 0 & -1 \\
0 & 0 & 0 & 0 & -1 & -1 & 0 & 0 & 0 & 0 & 4 & 0 & -1 & -1 & 0 \\
-1 & 0 & 0 & 0 & 0 & -1 & 0 & 0 & 0 & 0 & 0 & 4 & 0 & -1 & -1 \\
0 & 0 & 0 & 0 & 0 & 0 & -1 & -1 & 0 & -1 & -1 & 0 & 4 & 0 & 0 \\
0 & 0 & 0 & 0 & 0 & 0 & 0 & -1 & -1 & 0 & -1 & -1 & 0 & 4 & 0 \\
0 & 0 & 0 & 0 & 0 & 0 & -1 & 0 & -1 & -1 & 0 & -1 & 0 & 0 & 4
\end{pmatrix}
\begin{pmatrix}
c_1 \\ c_1 e^{-in\pi/3} \\ c_1 e^{-2in\pi/3} \\ c_1 e^{-in\pi} \\ c_1 e^{-4in\pi/3} \\ c_1 e^{-5in\pi/3} \\ c_2 \\ c_2 e^{-in\pi/3} \\ c_2 e^{-2in\pi/3} \\ c_2 e^{-in\pi} \\ c_2 e^{-4in\pi/3} \\ c_2 e^{-5in\pi/3} \\ c_3 \\ c_3 e^{-in\pi/3} \\ c_3 e^{-2in\pi/3}
\end{pmatrix}
= \frac{\xi_n}{\alpha^{(6)}}
\begin{pmatrix}
c_1 \\ c_1 e^{-in\pi/3} \\ c_1 e^{-2in\pi/3} \\ c_1 e^{-in\pi} \\ c_1 e^{-4in\pi/3} \\ c_1 e^{-5in\pi/3} \\ c_2 \\ c_2 e^{-in\pi/3} \\ c_2 e^{-2in\pi/3} \\ c_2 e^{-in\pi} \\ c_2 e^{-4in\pi/3} \\ c_2 e^{-5in\pi/3} \\ c_3 \\ c_3 e^{-in\pi/3} \\ c_3 e^{-2in\pi/3}
\end{pmatrix}.
\tag{C.1}
$$

For the case of 3+3 SU(2) fermions, the $V$ matrix reads

$$
\frac{V}{\alpha^{(6)}} =
\begin{pmatrix}
2 & 0 & 0 & 0 & 0 & 0 & -1 & 0 & 0 & 0 & 0 & 0 & 0 & -1 & 0 & 0 & 0 & 0 & 0 & 0 \\
0 & 2 & 0 & 0 & 0 & 0 & 0 & -1 & 0 & 0 & 0 & 0 & 0 & 0 & -1 & 0 & 0 & 0 & 0 & 0 \\
0 & 0 & 2 & 0 & 0 & 0 & 0 & 0 & -1 & 0 & 0 & 0 & 0 & 0 & 0 & -1 & 0 & 0 & 0 & 0 \\
0 & 0 & 0 & 2 & 0 & 0 & 0 & 0 & 0 & -1 & 0 & 0 & 0 & 0 & 0 & 0 & -1 & 0 & 0 & 0 \\
0 & 0 & 0 & 0 & 2 & 0 & 0 & 0 & 0 & 0 & -1 & 0 & 0 & 0 & 0 & 0 & 0 & -1 & 0 & 0 \\
0 & 0 & 0 & 0 & 0 & 2 & 0 & 0 & 0 & 0 & 0 & -1 & -1 & 0 & 0 & 0 & 0 & 0 & 0 & 0 \\
-1 & 0 & 0 & 0 & 0 & 0 & 4 & 0 & 0 & 0 & 0 & 0 & -1 & 0 & -1 & 0 & 0 & 0 & 0 & -1 \\
0 & -1 & 0 & 0 & 0 & 0 & 0 & 4 & 0 & 0 & 0 & 0 & 0 & -1 & 0 & -1 & 0 & 0 & -1 & 0 \\
0 & 0 & -1 & 0 & 0 & 0 & 0 & 0 & 4 & 0 & 0 & 0 & 0 & 0 & -1 & 0 & -1 & 0 & 0 & -1 \\
0 & 0 & 0 & -1 & 0 & 0 & 0 & 0 & 0 & 4 & 0 & 0 & 0 & 0 & 0 & -1 & 0 & -1 & -1 & 0 \\
0 & 0 & 0 & 0 & -1 & 0 & 0 & 0 & 0 & 0 & 4 & 0 & -1 & 0 & 0 & 0 & -1 & 0 & 0 & -1 \\
0 & 0 & 0 & 0 & 0 & -1 & 0 & 0 & 0 & 0 & 0 & 4 & 0 & -1 & 0 & 0 & 0 & -1 & -1 & 0 \\
0 & 0 & 0 & 0 & 0 & -1 & -1 & 0 & 0 & 0 & -1 & 0 & 4 & 0 & 0 & 0 & 0 & 0 & -1 & 0 \\
-1 & 0 & 0 & 0 & 0 & 0 & 0 & -1 & 0 & 0 & 0 & -1 & 0 & 4 & 0 & 0 & 0 & 0 & 0 & -1 \\
0 & -1 & 0 & 0 & 0 & 0 & 0 & 0 & -1 & 0 & 0 & 0 & 0 & 0 & 4 & 0 & 0 & 0 & -1 & 0 \\
0 & 0 & -1 & 0 & 0 & 0 & 0 & -1 & 0 & -1 & 0 & 0 & 0 & 0 & 0 & 4 & 0 & 0 & 0 & -1 \\
0 & 0 & 0 & -1 & 0 & 0 & 0 & 0 & -1 & 0 & -1 & 0 & 0 & 0 & 0 & 0 & 4 & 0 & -1 & 0 \\
0 & 0 & 0 & 0 & -1 & 0 & 0 & 0 & 0 & -1 & 0 & -1 & 0 & 0 & 0 & 0 & 0 & 4 & 0 & -1 \\
0 & 0 & 0 & 0 & 0 & 0 & -1 & 0 & -1 & 0 & -1 & -1 & 0 & -1 & 0 & -1 & 0 & -1 & 6 & 0 \\
0 & 0 & 0 & 0 & 0 & 0 & -1 & 0 & -1 & 0 & -1 & 0 & 0 & -1 & 0 & -1 & 0 & -1 & 0 & 6
\end{pmatrix}.
\tag{C.2}
$$

For the case of 3+3 SU(2) bosons, the $V$ matrix reads

$$
\frac{V}{\alpha^{(6)}} =
\begin{pmatrix}
10 & 0 & 0 & 0 & 0 & 0 & 1 & 0 & 0 & 0 & 0 & 0 & 0 & 1 & 0 & 0 & 0 & 0 & 0 & 0 \\
0 & 10 & 0 & 0 & 0 & 0 & 0 & 1 & 0 & 0 & 0 & 0 & 0 & 0 & 1 & 0 & 0 & 0 & 0 & 0 \\
0 & 0 & 10 & 0 & 0 & 0 & 0 & 0 & 1 & 0 & 0 & 0 & 0 & 0 & 0 & 1 & 0 & 0 & 0 & 0 \\
0 & 0 & 0 & 10 & 0 & 0 & 0 & 0 & 0 & 1 & 0 & 0 & 0 & 0 & 0 & 0 & 1 & 0 & 0 & 0 \\
0 & 0 & 0 & 0 & 10 & 0 & 0 & 0 & 0 & 0 & 1 & 0 & 0 & 0 & 0 & 0 & 0 & 1 & 0 & 0 \\
0 & 0 & 0 & 0 & 0 & 10 & 0 & 0 & 0 & 0 & 0 & 1 & 1 & 0 & 0 & 0 & 0 & 0 & 0 & 0 \\
1 & 0 & 0 & 0 & 0 & 0 & 8 & 0 & 0 & 0 & 0 & 0 & 1 & 0 & 1 & 0 & 0 & 0 & 0 & 1 \\
0 & 1 & 0 & 0 & 0 & 0 & 0 & 8 & 0 & 0 & 0 & 0 & 0 & 1 & 0 & 1 & 0 & 0 & 1 & 0 \\
0 & 0 & 1 & 0 & 0 & 0 & 0 & 0 & 8 & 0 & 0 & 0 & 0 & 0 & 1 & 0 & 1 & 0 & 0 & 1 \\
0 & 0 & 0 & 1 & 0 & 0 & 0 & 0 & 0 & 8 & 0 & 0 & 0 & 0 & 0 & 1 & 0 & 1 & 1 & 0 \\
0 & 0 & 0 & 0 & 1 & 0 & 0 & 0 & 0 & 0 & 8 & 0 & 1 & 0 & 0 & 0 & 1 & 0 & 0 & 1 \\
0 & 0 & 0 & 0 & 0 & 1 & 0 & 0 & 0 & 0 & 0 & 8 & 0 & 1 & 0 & 0 & 0 & 1 & 1 & 0 \\
0 & 0 & 0 & 0 & 0 & 1 & 1 & 0 & 0 & 0 & 1 & 0 & 8 & 0 & 0 & 0 & 0 & 0 & 1 & 0 \\
1 & 0 & 0 & 0 & 0 & 0 & 0 & 1 & 0 & 0 & 0 & 1 & 0 & 8 & 0 & 0 & 0 & 0 & 0 & 1 \\
0 & 1 & 0 & 0 & 0 & 0 & 1 & 0 & 1 & 0 & 0 & 0 & 0 & 0 & 8 & 0 & 0 & 0 & 1 & 0 \\
0 & 0 & 1 & 0 & 0 & 0 & 0 & 1 & 0 & 1 & 0 & 0 & 0 & 0 & 0 & 8 & 0 & 0 & 0 & 1 \\
0 & 0 & 0 & 1 & 0 & 0 & 0 & 0 & 1 & 0 & 1 & 0 & 0 & 0 & 0 & 0 & 8 & 0 & 1 & 0 \\
0 & 0 & 0 & 0 & 1 & 0 & 0 & 0 & 0 & 1 & 0 & 1 & 0 & 0 & 0 & 0 & 0 & 8 & 0 & 1 \\
0 & 0 & 0 & 0 & 0 & 0 & 0 & 1 & 0 & 1 & 0 & 1 & 1 & 0 & 1 & 0 & 1 & 0 & 6 & 0 \\
0 & 0 & 0 & 0 & 0 & 0 & 1 & 0 & 1 & 0 & 1 & 0 & 0 & 1 & 0 & 1 & 0 & 1 & 0 & 6
\end{pmatrix}.
\tag{C.3}
$$

For the case of 3+3 SB bosons, the $V$ matrix reads

$$\frac{V}{\alpha^{(6)}} = \begin{pmatrix}
2 & 0 & 0 & 0 & 0 & 0 & 1 & 0 & 0 & 0 & 0 & 0 & 0 & 1 & 0 & 0 & 0 & 0 & 0 & 0 \\
0 & 2 & 0 & 0 & 0 & 0 & 0 & 1 & 0 & 0 & 0 & 0 & 0 & 0 & 1 & 0 & 0 & 0 & 0 & 0 \\
0 & 0 & 2 & 0 & 0 & 0 & 0 & 0 & 1 & 0 & 0 & 0 & 0 & 0 & 0 & 1 & 0 & 0 & 0 & 0 \\
0 & 0 & 0 & 2 & 0 & 0 & 0 & 0 & 0 & 1 & 0 & 0 & 0 & 0 & 0 & 0 & 1 & 0 & 0 & 0 \\
0 & 0 & 0 & 0 & 2 & 0 & 0 & 0 & 0 & 0 & 1 & 0 & 0 & 0 & 0 & 0 & 0 & 1 & 0 & 0 \\
0 & 0 & 0 & 0 & 0 & 2 & 0 & 0 & 0 & 0 & 0 & 1 & 1 & 0 & 0 & 0 & 0 & 0 & 0 & 0 \\
1 & 0 & 0 & 0 & 0 & 0 & 4 & 0 & 0 & 0 & 0 & 0 & 1 & 0 & 1 & 0 & 0 & 0 & 0 & 1 \\
0 & 1 & 0 & 0 & 0 & 0 & 0 & 4 & 0 & 0 & 0 & 0 & 0 & 1 & 0 & 1 & 0 & 0 & 1 & 0 \\
0 & 0 & 1 & 0 & 0 & 0 & 0 & 0 & 4 & 0 & 0 & 0 & 0 & 0 & 1 & 0 & 1 & 0 & 0 & 1 \\
0 & 0 & 0 & 1 & 0 & 0 & 0 & 0 & 0 & 4 & 0 & 0 & 0 & 0 & 0 & 1 & 0 & 1 & 1 & 0 \\
0 & 0 & 0 & 0 & 1 & 0 & 0 & 0 & 0 & 0 & 4 & 0 & 1 & 0 & 0 & 0 & 1 & 0 & 0 & 1 \\
0 & 0 & 0 & 0 & 0 & 1 & 0 & 0 & 0 & 0 & 0 & 4 & 0 & 1 & 0 & 0 & 0 & 1 & 1 & 0 \\
0 & 0 & 0 & 0 & 0 & 1 & 1 & 0 & 0 & 0 & 1 & 0 & 4 & 0 & 0 & 0 & 0 & 0 & 1 & 0 \\
1 & 0 & 0 & 0 & 0 & 0 & 0 & 1 & 0 & 0 & 0 & 1 & 0 & 4 & 0 & 0 & 0 & 0 & 0 & 1 \\
0 & 1 & 0 & 0 & 0 & 0 & 1 & 0 & 1 & 0 & 0 & 0 & 0 & 0 & 4 & 0 & 0 & 0 & 1 & 0 \\
0 & 0 & 1 & 0 & 0 & 0 & 0 & 1 & 0 & 1 & 0 & 0 & 0 & 0 & 0 & 4 & 0 & 0 & 0 & 1 \\
0 & 0 & 0 & 1 & 0 & 0 & 0 & 0 & 1 & 0 & 1 & 0 & 0 & 0 & 0 & 0 & 4 & 0 & 1 & 0 \\
0 & 0 & 0 & 0 & 1 & 0 & 0 & 0 & 0 & 1 & 0 & 1 & 0 & 0 & 0 & 0 & 0 & 4 & 0 & 1 \\
0 & 0 & 0 & 0 & 0 & 0 & 0 & 1 & 0 & 1 & 0 & 1 & 1 & 0 & 1 & 0 & 1 & 0 & 6 & 0 \\
0 & 0 & 0 & 0 & 0 & 0 & 1 & 0 & 1 & 0 & 1 & 0 & 0 & 1 & 0 & 1 & 0 & 1 & 0 & 6
\end{pmatrix}. \tag{C.4}$$

The contact matrix for 3 SU(3) fermions reads

$$\frac{V}{\alpha^{(3)}} = \begin{pmatrix}
3 & 0 & 0 & -1 & -1 & -1 \\
0 & 3 & 0 & -1 & -1 & -1 \\
0 & 0 & 3 & -1 & -1 & -1 \\
-1 & -1 & -1 & 3 & 0 & 0 \\
-1 & -1 & -1 & 0 & 3 & 0 \\
-1 & -1 & -1 & 0 & 0 & 3
\end{pmatrix}. \tag{C.5}$$

# D  Energy correction to order $\mathcal{O}\left(\frac{1}{g}\right)$: Bethe Ansatz derivation

In this section, we explicitly derive the energy correction at large but finite interaction strength $g$ for a $SU(2)$ mixture of $N$ particles. To do so, we consider the first-order expansion of the Bethe equations close to the limit $\frac{1}{g} \to 0$. In the following, we consider the rescaled interaction strength $u \doteq \frac{2m}{\hbar^2} g$, so that $u$ has the dimension of a wavevector. We start from the Bethe equations valid for any interaction strength, which read:

$$\begin{cases}
L k_j = 2\pi \mathcal{I}_j + (-1)^{\eta_B} \sum_{b=1}^{N_\downarrow} 2\arctan\left(\frac{2(\tilde{\lambda}_b - k_j)}{u}\right) + \eta_B \sum_{\ell=1}^{N} 2\arctan\left(\frac{k_\ell - k_j}{u}\right), \\
\sum_{j=1}^{N} 2\arctan\left(\frac{2(\tilde{\lambda}_a - k_j)}{u}\right) = 2\pi \mathcal{J}_a + \sum_{b=1}^{N_\downarrow} 2\arctan\left(\frac{\tilde{\lambda}_a - \tilde{\lambda}_b}{u}\right),
\end{cases} \tag{D.1}$$

where $\eta_B = 1$ for bosons and $\eta_B = 0$ for fermions. The quantum numbers $\mathcal{I}_j$ and $\mathcal{J}_m$ are defined in Section 2.3. In the limit $u \to \infty$, one has $\frac{\tilde{\lambda}_a}{u} \gg \frac{k_j}{u}$, therefore we expand the arcotangent function according to the first-order expansions $\arctan(a + x) = \arctan(a) + \frac{x}{1+a^2} + \mathcal{O}(x^2)$

and $\arctan(x) = x + \mathcal{O}(x^2)$. We introduce the rescaled spin rapidities $\Lambda_a \doteq \frac{2\tilde{\lambda}}{u}$ and obtain:

$$\begin{cases} Lk_j = 2\pi\mathcal{I}_j + (-1)^{\eta_B}\sum_{b=1}^{N_{\downarrow}} 2\arctan(\Lambda_b) - (-1)^{\eta_B}\frac{4}{u}k_j\sum_{b=1}^{N_{\downarrow}}\frac{1}{1+\Lambda_b^2} + \frac{2\eta_B}{u}(\sum_{\ell}k_{\ell} - Nk_j), \\[2ex] 2\arctan(\Lambda_a) = \frac{4}{uN}\sum_j k_j\frac{1}{1+\Lambda_a^2} + \frac{2\pi}{N}\mathcal{J}_a + \sum_{b=1}^{N_{\downarrow}}\frac{2}{N}\arctan\left(\frac{\Lambda_a - \Lambda_b}{2}\right). \end{cases}$$
(D.2)

A straightforward substitution of the right-hand-side of the second equation in the first equation yields:

$$\begin{cases} Lk_j = 2\pi\mathcal{I}_j + (-1)^{\eta_B}\frac{2\pi}{N}\sum_{b=1}^{N_{\downarrow}}\mathcal{J}_b + \frac{1}{u}\left(\frac{1}{N}\sum_{\ell}k_{\ell} - k_j\right)\left(2N\eta_B + (-1)^{\eta_B}\sum_{b=1}^{N_{\downarrow}}\frac{4}{1+\Lambda_b^2}\right), \\[2ex] 2\arctan(\Lambda_a) = \frac{1}{uN}\sum_j k_j\frac{4}{1+\Lambda_a^2} + \frac{2\pi}{N}\mathcal{J}_a + \sum_{b=1}^{N_{\downarrow}}\frac{2}{N}\arctan\left(\frac{\Lambda_a - \Lambda_b}{2}\right), \end{cases}$$
(D.3)

where we use $\sum_{a,b=1}^{N_{\downarrow}}\arctan\left(\frac{\Lambda_a - \Lambda_b}{2}\right) = 0$. In the limit $\frac{1}{u} \to 0$, one recovers Eqs.(7) and (9). We can write the first equation in a more compact form:

$$k_j = \frac{2\pi}{L}\mathcal{I}_j + \frac{\chi}{L} + \frac{1}{uL}\delta k_j,$$
(D.4)

where we defined:

$$\chi \doteq (-1)^{\eta_B}\frac{2\pi}{N}\sum_{b=1}^{N_{\downarrow}}\mathcal{J}_b,$$
(D.5)

$$\delta k_j \doteq \left(\frac{1}{N}\sum_{\ell}k_{\ell} - k_j\right)\left(2N\eta_B + (-1)^{\eta_B}\sum_{b=1}^{N_{\downarrow}}\frac{4}{1+\Lambda_b^2}\right).$$
(D.6)

Remarkably, $\chi$ can include any shift to the rapidities $k_j$ which does not depend on the index $j$, as for instance an artificial gauge field.

The energy in this strongly interacting limit is:

$$\begin{aligned} \frac{2m}{\hbar^2}E_{1/u} = \sum_j k_j^2 &= \sum_j\left(\frac{2\pi}{L}\mathcal{I}_j + \frac{\chi}{L} + \frac{1}{uL}\delta k_j\right)^2 \\ &= \sum_j\left(\frac{2\pi}{L}\mathcal{I}_j + \frac{\chi}{L}\right)^2 + \frac{2}{uL}\sum_j\left(\frac{2\pi}{L}\mathcal{I}_j + \frac{\chi}{L}\right)\delta k_j + \mathcal{O}\left(\frac{1}{u^2}\right) \doteq \frac{2m}{\hbar^2}\left(E_{\infty} + \delta E_{1/u} + \mathcal{O}\left(\frac{1}{u^2}\right)\right), \end{aligned}$$
(D.7)

where we introduced the energy correction:

$$\delta E_{1/u} = \frac{\hbar^2}{uLm}\sum_j\left(\frac{2\pi}{L}\mathcal{I}_j + \frac{\chi}{L}\right)\delta k_j.$$
(D.8)

Table 7: Energy corrections to first order in $1/u$ for $N = 4$ and $N_\downarrow = 2$ bosons. The spin rapidities are calculated from Eq. (9). The results are in units of $J_{\text{eff}}$.

| $\Lambda_1$ | $\Lambda_2$ | $-\dfrac{\delta E_{1/u}}{J_{\text{eff}}}$ |
|:---:|:---:|:---:|
| $1/\sqrt{3}$ | $-1/\sqrt{3}$ | 2 |
| 0 | $\infty$ | 4 |
| 1 | $\infty$ | 6 |
| $-1$ | $\infty$ | 6 |
| $i$ | $-i$ | 6 |
| $\infty$ | $-\infty$ | 8 |

Neglecting the $\mathcal{O}\left(\frac{1}{u^2}\right)$ terms, the energy correction reads:

$$
\begin{aligned}
\delta E_{1/u} &= \frac{\hbar^2}{uLm} \sum_j \left(\frac{2\pi}{L}\mathcal{I}_j + \frac{\chi}{L}\right)\left(\frac{1}{N}\sum_\ell k_\ell - k_j\right)\left(2N\eta_B + (-1)^{\eta_B}\sum_{b=1}^{N_\downarrow}\frac{4}{1+\Lambda_b^2}\right) \\
&= \frac{\hbar^2}{uLm}\left(\frac{1}{N}\left(\frac{2\pi}{L}\sum_\ell \mathcal{I}_\ell + \frac{N\chi}{L}\right)^2 - \sum_j\left(\frac{2\pi}{L}\mathcal{I}_j + \frac{\chi}{L}\right)^2\right)\left(2N\eta_B + (-1)^{\eta_B}\sum_{b=1}^{N_\downarrow}\frac{4}{1+\Lambda_b^2}\right) \\
&= -\frac{\hbar^2}{uLm}\left(\frac{4\pi^2}{L^2}\sum_j \mathcal{I}_j^2 - \frac{4\pi^2}{NL^2}\left(\sum_\ell \mathcal{I}_\ell\right)^2\right)\left(2N\eta_B + (-1)^{\eta_B}\sum_{b=1}^{N_\downarrow}\frac{4}{1+\Lambda_b^2}\right).
\end{aligned}
\tag{D.9}
$$

In order to have only first-order correction in $\frac{1}{u}$, the spin rapidities $\Lambda_b$ are obtained solving the corresponding Bethe equation in the limit $\frac{1}{u} \to 0$, Eq. (9). From the last line of Eq. (D.9), we see that the center of mass momentum does not contribute to the first-order corrections to the total energy. Moreover, if we introduce an effective coupling

$$
J_{\text{eff}} = \frac{\hbar^2}{uLm}\left(\frac{4\pi^2}{L^2}\sum_j \mathcal{I}_j^2 - \frac{4\pi^2}{NL^2}\left(\sum_\ell \mathcal{I}_\ell\right)^2\right),
\tag{D.10}
$$

the correction to the energy can be expressed as:

$$
\delta E_{1/u} = -J_{\text{eff}}\left(2N\eta_B + (-1)^{\eta_B}\sum_{b=1}^{N_\downarrow}\frac{4}{1+\Lambda_b^2}\right),
\tag{D.11}
$$

which, up to a constant shift in the bosonic case, coincides with the Bethe Ansatz solution for the energy of an isotropic Heisenberg spin chain. The ferromagnetic or antiferromagnetic nature of the ground state is determined by the value of $\eta_B$ and therefore by the statistics of the mixture. Remark that $J_{\text{eff}}uL/2$ is equal to the difference between the total kinetic energy and the center-of-mass kinetic energy, and that $J_{\text{eff}}uL/2 = J_{\text{eff}}\,gmL/\hbar^2 = mL/\hbar^2\,\alpha^{(N)}$.

We computed explicitly the energy correction in the case of $N = 4$ and $N_\downarrow = 2$ bosons. First, to determine the correct $\Lambda_1$ and $\Lambda_2$, we solved the Bethe equations (9). Then, we used Eq. (D.11) to evaluate the first-order correction to the energy at infinite interactions. Since Eq. (D.11) diverges for $\Lambda_{1,2} = \pm i$, in order to obtain the corresponding correction we had to introduce the regularization $\Lambda_{1,2} = \epsilon \pm i$ and then compute the limit of Eq. (D.11) for $\epsilon \to 0$.

We show the results in Table 7. The energy corrections coincide with the results presented in the second column of Table 1 ($-\frac{\delta E_{1/u}}{J_{\text{eff}}} = \tilde{\xi}_n$), where we computed the first-order energy correction using the necklace Ansatz. More details on the Bethe Ansatz solution for $N = 4$ and $N_\downarrow = 2$, including the exact expression of the corresponding eigenstates, are available in [64].

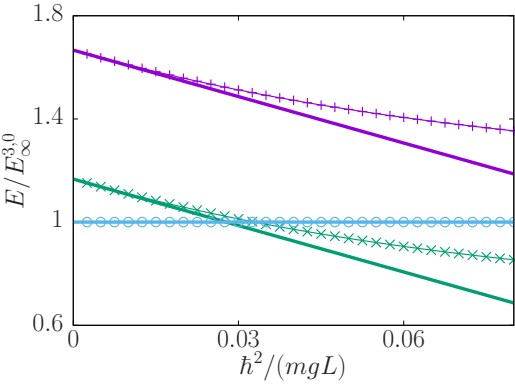

Figure 4: First three energy levels for a system of 2+1 fermions, in units of $E^{3,0} = 4\pi^2\hbar^2/(mL^2)$ as functions of $1/g$ (in units of $mL/\hbar^2$). The horizontal blue line corresponds to a fully antisymmetric state, and the necklace solution ($n = 0$ in Table 8) coincides with the Bethe Ansatz one for any interaction strength (blue circles). The cross and the plus symbols (the lines are guides to the eyes) lines are the Bethe solutions that correspond to the first two excited states at infinite interactions. The tangent thick lines correspond to the necklace solution respectively for $n = 1$ (green line) and $n = 2$ (violet line).

# E  Analysis of the strongly interacting limit

With the aim to specify the validity range of the strong-interaction expansion used in Sec. D and in the necklace Ansatz, we compare the solution obtained in the strongly interacting limit for a system of 2+1 fermions with that obtained by the Bethe equations for intermediate interactions given in Eq. (D.1).

In Fig. 4, we plot the first three energy levels of the Bethe Ansatz solution as a function of the inverse of the interaction strength. The ground state at infinite interactions corresponds to the fully antisymmetric state (see Table 8). It does not depend on the interaction strength (circles) and coincides with the necklace solution with $n = 0$ (horizontal blue line). The two excited states at $1/g = 0$ (cross and plus symbols) correspond to the necklace solutions (tangent thick green and violet lines) with $n = 1$ and $n = 2$ respectively.

We observe that the solutions obtained in the strong-interaction limit agree with the energies from the Bethe Ansatz equations for $1/g \lesssim 0.03 mL/\hbar^2$. This corresponds to an effective interaction parameter $\hbar^2 mLg/N = 10$, which provides an estimate for the ratio between the interaction and kinetic energies for which the necklace Ansatz can still produce accurate results.

Table 8: Solution for the fermionic 2+1 system. $\tilde{\xi}_n$ are the rescaled eigenvalues $\xi_n/\tilde{\alpha}^{(3)}$. $\gamma^{(2)}$ indicates the symmetry of the solution, and the associated Young diagram. The last column corresponds to the energies at infinite interactions (in units of $\hbar^2/(mL^2)$).

| $n$ | $\tilde{\xi}_n$ | $c_1$ | $\gamma^{(2)}$ | YD | $E_\infty^{3,n}$ |
|---|---|---|---|---|---|
| 0 | 0 | 1 | -3 | ⊟ | $4\pi^2$ |
| 1 | 3 | 1 | 0 | ⊞ | $\dfrac{14\pi^2}{3}$ |
| 2 | 3 | 1 | 0 | ⊞ | $\dfrac{20\pi^2}{3}$ |

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
