# Peer review of "Necklace Ansatz for strongly repulsive spin mixtures on a ring"

_SciPost Physics Core, doi:SciPost Phys. Core 8, 022 (2025)_

## Round 1 · Referee Report · Anonymous (Referee 2) · 2024-11-15

Report
In the work ``Necklace Ansatz for strongly repulsive spin mixtures on a ring''
the authors propose an ansatz which reminds of perturbation theory
and apply it to a strong (infinite) repulsive regime for SU(2) and SU(3) cases.
This strongly reminds of the situation of Ogata and Shiba which exactly describe the situation for the SU(2)
in this limit. There is also a work which builds on top of their work precisely on SU(N) extensions.
Both works are not at all mentioned in this manuscript since these are basic examples
where the authors could test the strength of the method proposed.
Here there is a contact matrix that has to be elaborated for each single case
and which seems the interaction transported into the new basis of the problem which is
given by necklaces and snippets.
It is shown in few examples that choosing this basis is a notable reduction in states to be considered
and this would be more pronounced for larger system sizes.
The end-product, however, is a system of largely overdetermined equations for the parameters which describe the infinitely system.
And it does not help to just state that it seems just to be overdetermined and referring to some gauge invariance.
It is there and the authors apparently do not have a proper method to deal with this problem.
Seeing the slightly small system sizes given as examples (up to SU(3) 3 particles) one is tempted to
a) see what is written in the work on the extension of Ogata and Shiba and compare with that work.
b) ask the authors where they think to arrive with their ansatz in terms of number of sites, particles, kappa?
This work is interesting for this infinitely interacting regime but elsewhere not applicable.
Therefore it is a rather niche application than a work of broader interest and should appear elsewhere.
Consequently, it should not be published in SciPost Physics.
Remarks and questions:
The article (beginning with the title) speaks of boson-fermi mixtures to be addressed, however I don't see where this is done.
At the moment every regarded system is just of purely bosons or of purely fermions,
and no mixture of the both has been considered.
In the abstract there is a ``repulsive'' missing, I guess.
For an even number of particles there is no k=0 value, right?
Equation (8) there is a symbol R: is it permutations as usual?
The authors speak of ``relative'' integers: what are they?
The different sums in (13),(14) should be marked otherwise; now it seems that two parameters areto be summed over.
The Young-tableaus are sometimes flipped in certain tables; it should be stated
a) that they appear flipped and hence the axix of (anti)-symmetrization are exchanged
b) the ``more exotic'' tableaus correspond to a different representation of the symmetric group S_n
c) standard young tableaus do not show the singlets in the states; for example the young tableau for SU(2)
would be composed only of the first (of two: anti-symmetrization) row.
I have a problem with formula (19) and values for \gamma^{(2)} in that it is not clear to me
whether the summation starts in 0 or 1, or wherever it starts; I cannot reproduce the values of the various Young tableaus in either case.
I have looked into Ref. [62] and could not get an answer to my question.
According to Table 1 it could seem that it is the spin/casimir operator, but it is probably not like that.
Where (in which chapter or equation) in the books Refs. [60],[61] is this found?
Please give with books also at least the corresponding chapter as reference.
the authors propose an ansatz which reminds of perturbation theory
and apply it to a strong (infinite) repulsive regime for SU(2) and SU(3) cases.
This strongly reminds of the situation of Ogata and Shiba which exactly describe the situation for the SU(2)
in this limit. There is also a work which builds on top of their work precisely on SU(N) extensions.
Both works are not at all mentioned in this manuscript since these are basic examples
where the authors could test the strength of the method proposed.
Here there is a contact matrix that has to be elaborated for each single case
and which seems the interaction transported into the new basis of the problem which is
given by necklaces and snippets.
It is shown in few examples that choosing this basis is a notable reduction in states to be considered
and this would be more pronounced for larger system sizes.
The end-product, however, is a system of largely overdetermined equations for the parameters which describe the infinitely system.
And it does not help to just state that it seems just to be overdetermined and referring to some gauge invariance.
It is there and the authors apparently do not have a proper method to deal with this problem.
Seeing the slightly small system sizes given as examples (up to SU(3) 3 particles) one is tempted to
a) see what is written in the work on the extension of Ogata and Shiba and compare with that work.
b) ask the authors where they think to arrive with their ansatz in terms of number of sites, particles, kappa?
This work is interesting for this infinitely interacting regime but elsewhere not applicable.
Therefore it is a rather niche application than a work of broader interest and should appear elsewhere.
Consequently, it should not be published in SciPost Physics.
Remarks and questions:
The article (beginning with the title) speaks of boson-fermi mixtures to be addressed, however I don't see where this is done.
At the moment every regarded system is just of purely bosons or of purely fermions,
and no mixture of the both has been considered.
In the abstract there is a ``repulsive'' missing, I guess.
For an even number of particles there is no k=0 value, right?
Equation (8) there is a symbol R: is it permutations as usual?
The authors speak of ``relative'' integers: what are they?
The different sums in (13),(14) should be marked otherwise; now it seems that two parameters areto be summed over.
The Young-tableaus are sometimes flipped in certain tables; it should be stated
a) that they appear flipped and hence the axix of (anti)-symmetrization are exchanged
b) the ``more exotic'' tableaus correspond to a different representation of the symmetric group S_n
c) standard young tableaus do not show the singlets in the states; for example the young tableau for SU(2)
would be composed only of the first (of two: anti-symmetrization) row.
I have a problem with formula (19) and values for \gamma^{(2)} in that it is not clear to me
whether the summation starts in 0 or 1, or wherever it starts; I cannot reproduce the values of the various Young tableaus in either case.
I have looked into Ref. [62] and could not get an answer to my question.
According to Table 1 it could seem that it is the spin/casimir operator, but it is probably not like that.
Where (in which chapter or equation) in the books Refs. [60],[61] is this found?
Please give with books also at least the corresponding chapter as reference.
Recommendation
Reject
Strengths
- New approach.
- Interesting physical system.
- Highly non-trivial many-body quantum effects.
Weaknesses
- Very technical.
- At the end the method requires heavy numerical calculations.
Report
The authors develop an interesting ansatz to solve the problem of a mixture of strongly-interacting fermions in a ring. The physical systems is surely relevant for the current experiments in low-dimensional quantum gases.
Their "necklace ansatz" seems somehow better with respect to the familiar Bethe ansatz. In the case of a large number of fermions one has to face highly numerical calculations which, however, involve an Hilbert space with a reduced dimension.
The paper is surely interesting and accurately written. However, I do not see exciting new physical results. Maybe Sci Post Physics Core could be a more appropriate destination for this paper.
Their "necklace ansatz" seems somehow better with respect to the familiar Bethe ansatz. In the case of a large number of fermions one has to face highly numerical calculations which, however, involve an Hilbert space with a reduced dimension.
The paper is surely interesting and accurately written. However, I do not see exciting new physical results. Maybe Sci Post Physics Core could be a more appropriate destination for this paper.
Requested changes
None.
Recommendation
Accept in alternative Journal (see Report)

---

## Round 2 · Author Response

Dear Editor,
We appreciate your handling of our submission and the opportunity to revise and resubmit the manuscript. Further, we would like to thank the Referees for taking the time to review our manuscript and for their insightful comments. We believe that the potential impact of our work has been misjudged and would like to stress once more that the necklace Ansatz (i) allows one to simplify drastically - with respect to the Bethe Ansatz - calculations of the eigenstates for strongly repulsive fermionic (or bosonic) mixtures on a ring, (ii) provides a complete basis of solutions, (iii) gives a physical insight into the relation between the spin structure of the solution and the total momentum. These considerations led us to submit our work to SciPost Physics, as we strongly believe it aligns well with the journal’s scope and standards. Nevertheless, following Editor’s recommendation, we thereby resubmit a revised version of our manuscript Necklace Ansatz for strongly repulsive spin mixtures on a ring to SciPost Physics Core. In the following, we provide a point-to-point answers to all the comments raised by the Referees. Yours Sincerely, The Authors
- Response to Referee 1
1) The authors develop an interesting ansatz to solve the problem of a mixture of strongly-interacting fermions in a ring. The physical systems is surely relevant for the current experiments in low-dimensional quantum gases. Their "necklace ansatz" seems somehow better with respect to the familiar Bethe ansatz. In the case of a large number of fermions one has to face highly numerical calculations which, however, involve an Hilbert space with a reduced dimension. The paper is surely interesting and accurately written. However, I do not see exciting new physical results.
Reply: We thank Referee 1 for taking the time to review our work and for finding the paper ‘interesting and accurately written’. We emphasize that our method involves numerical calculations that are computationally less intensive compared to the Bethe ansatz, while still granting access to the exact wavefunction of the system. Further, our approach yields a complete set of solutions at strong coupling, which can be a noticeably harder task if the Bethe Ansatz methods are used instead. The main breakthrough of our work is on the methodological side, and we highlight that this is a major step in the strong-coupling solutions. Concerning the new physical results, the necklace solution provides a physical interpretation of the solution that allows one to connect the spin states to the total momentum. This is the second breakthrough of our work. Moreover we provide an analytical prediction of the energy landscape and persistent currents for large but finite interactions. Our results highlight that the energy landscape is only partially split to order 1/g and may provide a strict benchmark of numerical simulations.
- Response to Referee 2
Referee: In the work Necklace Ansatz for strongly repulsive spin mixtures on a ring” the authors propose an ansatz which reminds of perturbation theory and apply it to a strong (infinite) repulsive regime for SU(2) and SU(3) cases. This strongly reminds of the situation of Ogata and Shiba which exactly describe the situation for the SU(2) in this limit. There is also a work which builds on top of their work precisely on SU(N) extensions. Both works are not at all mentioned in this manuscript since these are basic examples where the authors could test the strength of the method proposed.
Our reply: At the outset, we thank the Referee for taking the time to review our work. Below, we provide a point-by-point reply to the questions and comments that we identified in the report.
• 1) Seeing the slightly small system sizes given as examples (up to SU(3) 3 particles) one is tempted to a) see what is written in the work on the extension of Ogata and Shiba and compare with that work. b) ask the authors where they think to arrive with their ansatz in terms of number of sites, particles, kappa?
Reply: a) The work of Ogata and Shiba concerns a SU(2) strongly repulsive fermionic mixture on a lattice. Our work concerns SU(κ) fermionic (or bosonic) mixtures on the continuum. Furthermore, Ogata and Shiba do not propose a method to evaluate the many-body wavefunction, but just develop a method to calculate the momentum distribution. In this sense, our work has a different focus than the Ogata-Shiba one. Indeed, as an extension of the current work it might be possible to formulate a similar necklace Ansatz for fermionic mixtures on the lattice (while bosonic mixtures are not integrable on the lattice, as neither the Bose-Hubbard model is). It is an interesting suggestion, however, in order to keep this work sufficiently focused, we prefer to postpone this analysis for a later work. We added the citation to Ogata and Shiba as reference [53] in the new version of the manuscript.
b) We thank the referee for this question. As we have already pointed out, we are studying a homogeneous ring, so in our system there are not physical sites. The system at strong coupling is mapped to a spin chain where the sites correspond to the particles. The number of particles and components (κ) determined the number of snippets Ns, namely the size of the contact matrix, whose diagonalization is not a limiting factor in our procedure. The step that is more time demanding is the calculation of the matrix elements because we need to calculate Ns(Ns − 1)/2 elements (half of the off-diagonal ones) by comparing the position of N spin in pairs of snippets and Ns elements (the diagonal ones) by checking the neighbors of each spin in the snippet. In principle, this could be further reduced, up to Ns*Ns /4, by taking advantage of the parity symmetry. In this work we have provided some examples with a relative small number of particle in order to (i) have, where possible, the comparison with the Bethe Ansatz solution and previous works in the literature, and (ii) provide a pedagogical illustration of the method. Since the method is close in spirit to the trapped case, where all the states have been calculated for N = 16 and κ = 2 (Ns = 12860) [Phys. Rev. A 94, 023606 (2016)] we estimate that one could arrive to a similar numbers of snippets in the current case.
• 2) The article (beginning with the title) speaks of boson-fermi mixtures to be addressed, however I don’t see where this is done. At the moment every regarded system is just of purely bosons or of purely fermions, and no mixture of the both has been considered.
Reply: The term "mixtures" in the title refers to spin mixtures, which indicates a mixture of different spin flavors. Similarly, during the text the term "mixtures" refers to fermionic or bosonic spin mixtures. Moreover we have specified in several places, starting from the abstract, that we are considering fermionic or bosonic mixtures.
• 3) In the abstract there is a repulsive” missing, I guess.
Reply: We thank the Referee for pointing this out, we specified in the abstract that we focus on repulsive strong interactions.
• 4) For an even number of particles there is no k = 0 value, right?
Reply: The values of the charge rapidities here correspond to the ones of the non-interacting gas. In this case, for fermionic systems, the value of k = 0 is allowed both for even and odd number of particles. However, the rapidity k = 0 is not permitted by the quantum numbers of Bose gases with an even number of particles. This difference leads to the parity effect observed in fermionic systems, which is absent in bosonic systems. Specifically, the total momentum P = ℏ ∑j kj of a non-interacting Fermi gas can only be zero when the particle number is odd. In contrast, for bosonic systems, this value is permitted regardless of the parity of the particle number N (for a more exhaustive treatment, see A.J.Leggett, in Granular Nanoelectronics, edited by D. K.Ferry, J. R. Barker, and C. Jacoboni (Springer, US, Boston, 1991), pp. 297–311)
• 5) Equation (8) there is a symbol R: is it permutations as usual?
Reply: As correctly pointed out by the Referee, the symbol R in Eq.(8) refers to the permutations. We clarify this point in the paragraph right after the equation.
• 6) The authors speak of relative” integers: what are they?
Reply: Relative integers are the set of positive and negative natural numbers, plus the zero. It is usually indicated with Z. The term ‘relative’ emphasizes that the integers include both positive and negative values (as opposed to just natural numbers, which are typically non-negative).
• 7) The different sums in (13), (14) should be marked otherwise; now it seems that two parameters are to be summed over.
Reply: We thank the Referee for this comments, it is well-taken. We modified the notation of the two summations in Eq.(13) and (14) to remove the ambiguity between the type of summation and the indexes which are to be summed over.
• 8) The Young-tableaus are sometimes flipped in certain tables; it should be stated a) that they appear flipped and hence the axix of (anti)-symmetrization are exchanged b) the more exotic” tableaus correspond to a different representation of the symmetric group Sn c) standard young tableaus do not show the singlets in the states; for example the young tableau for SU(2) would be composed only of the first (of two: anti-symmetrization) row.
Reply: Please note that (a) In the paragraph between Eq. (18) and Eq. (19) it is specified that horizontal diagrams correspond to fully symmetric states and vertical diagrams to full anti-symmetric states. (b) In the same paragraph we only presented Young’s diagrams as representations of SU (κ) to lighten the discussion. We agree with the Referee that this is indeed misleading, so we added one sentence, at the beginning of the paragraph, to state that they constitute equivalently a representation of Sn and a second sentence just after to clarify that each diagram corresponds to a different representation, as requested by the referee. (c) The singlet corresponds to a single column (see point (a)).
• 9) I have a problem with formula (19) and values for γ(2) in that it is not clear to me whether the summation starts in 0 or 1, or wherever it starts; I cannot reproduce the values of the various Young tableaus in either case. I have looked into Ref. [62] and could not get an answer to my question. According to Table 1 it could seem that it is the spin/casimir operator, but it is probably not like that.
Reply: As it is written in the manuscript, μi is the number of boxes at the line i, so the sum start from 1 (the first row) up to the last row. As the referee has pointed out, and as it was written in the manuscript, γ(2) are the eigenvalues of the Casimir operator Γ(2). In order to clarify the sentence above Eq. (19), we have replaced "line" with "row".
• 10) Where (in which chapter or equation) in the books Refs. [60],[61] is this found? Please give with books also at least the corresponding chapter as reference.
Reply: We thank the referee for their question. Indeed the most suitable reference for the explicit definition of the Γ(2) is the work of Akiva Novoselsky and Jacob Katriel [Phys. Rev. A 49, 833 (1994)]. We have added this citation, added the chapter for the book of James and Liebeck and canceled the citation to the book of James and Kerber.

---

## Round 2 · List of Changes

List of changes (in blue in the manuscript)
- we have added the word "repulsive"in the abstract
- At line 141 we have added the sentence "The summation runs over all the possible permutations $R$ of the $N_2$ spin rapidities."
- We have changed the notations in Eqs. (13) and (14) and the new notations have been explained at lines 210 and 211.
- At line 256, we have added " (or equivalently SN )"
- At lines 261/262, we have added ", each corresponding to a different representation of SN "
- At line 267 we have replaced the word "line" with "row".
- We have added references [53], and [61].
- We have added the chapter in Ref. [62], and canceled the citation to the book of James and Kerber.

---

## Editorial Decision

published